# On Quantizing Neural Representation for Variable-Rate Video Coding

**Junqi Shi, Zhujia Chen, Hanfei Li, Qi Zhao, Ming Lu,**[*] **Tong Chen, Zhan Ma**

School of Electronic Science and Engineering, Nanjing University
{junqishi,zhujiachen,hanfei_li,qizhao}@smail.nju.edu.cn,
{minglu,chentong,mazhan}@nju.edu.cn

## Abstract

This work introduces NeuroQuant, a novel post-training quantization (PTQ) approach tailored to non-generalized Implicit Neural Representations for variable-rate Video Coding (INR-VC). Unlike existing methods that require extensive weight retraining for each target bitrate, we hypothesize that variable-rate coding can be achieved by adjusting quantization parameters (QPs) of pre-trained weights. Our study reveals that traditional quantization methods, which assume inter-layer independence, are ineffective for non-generalized INR-VC models due to significant dependencies across layers. To address this, we redefine variable-rate INR-VC as a mixed-precision quantization problem and establish a theoretical framework for sensitivity criteria aimed at simplified, fine-grained rate control. Additionally, we propose network-wise calibration and channel-wise quantization strategies to minimize quantization-induced errors, arriving at a unified formula for representation-oriented PTQ calibration. Our experimental evaluations demonstrate that NeuroQuant significantly outperforms existing techniques in varying bitwidth quantization and compression efficiency, accelerating encoding by up to eight times and enabling quantization down to INT2 with minimal reconstruction loss. This work introduces variable-rate INR-VC for the first time and lays a theoretical foundation for future research in rate-distortion optimization, advancing the field of video coding technology. The materials will be available at https://github.com/Eric-qi/NeuroQuant.

## 1 Introduction

Implicit Neural Representations (INRs) (Sitzmann et al., 2020; Chen et al., 2021a) have recently introduced a new approach to video coding. They focus on learning a mapping from coordinates, like frame indices, to pixel values, such as colors. This represents a significant departure from the widely used variational autoencoder (VAE)-based frameworks (Lu et al., 2019; Li et al., 2021a; Lu et al., 2024), which rely on generalized models trained on large datasets to create compact representations for various input signals. Instead, INR-based video coding (INR-VC) encodes each video as a unique neural network through end-to-end training, removing the need for extensive datasets. By using specific, non-generalized network weights for each video, INR-VC provides a tailored video coding method that has shown promising results (Chen et al., 2023; Kwan et al., 2024a).

INR-VC typically focuses on two main objectives: 1) **Representation**, where a neural network models the target video with a minimized distortion, and 2) **Compression**, where the network's weights are compressed to lower the bitrate. Many prominent methods adopt a consistent precision (quantization bitwidth) for all weights before lossless entropy coding, meaning the video bitrate depends solely on the number of learnable weights. Consequently, independent weight training is needed for each target bitrate, making the process very time-consuming. For example, encoding a 1080p video with 600 frames at a specific bitrate can take up to 10 hours.

To address this inefficiency, we consider how bitrate is managed in pretrained INR-VC model, which is proportional to the sum of the bitwidth of each weight. Inspired by generalized codecs (Sullivan

---

[*]Corresponding Author

Figure 1: **Left**: Typical INR-VCs assume a consistent bitwidth and require separate weight training with varying quantities for each target rate. **Right**: The proposed NeuroQuant achieves variable rate by modifying the corresponding QPs, significantly reducing training costs.

et al., 2012; Li et al., 2023) that adjust quantization parameters (QPs) (Wang & Kwong, 2008) to control bitrate, we pose the hypothesis: *Can variable-rate INR-VC be achieved by modifying the QP of post-training weights*, thus eliminating the need for repeated model training for each target rate? In the context of weight quantization, this can be approached by: 1) allocating quantization bitwidth to match the target bitrate, and 2) calibrating QPs to preserve reconstruction fidelity.

However, directly adopting a consistent quantization bitwidth cannot support fine-grained rate control, e.g., only seven options from INT2 to INT8 are available. Additionally, existing mixed-precision quantization methods (Nagel et al., 2021; Chen et al., 2021b), primarily designed for general-purpose neural networks, encounter two key problems when applied to non-generalized INR-VCs. First, mixed-precision algorithms (Dong et al., 2019; 2020; Chen et al., 2021b) typically assume inter-layer independence with tolerable approximation errors. This assumption breaks down in non-generalized INR-VCs, where layers exhibit significant dependencies. Second, popular layer-wise calibration methods[1] (Nagel et al., 2020; Li et al., 2021b) also rely on inter-layer independence and aims at generalizing the network, making them unsuitable for INR-VC. Therefore, a dedicated quantization methodology tailored for variable-rate INR-VC is necessary.

In this work, we explore, for the first time, the post-training quantization (PTQ) of weights in non-generalized INR-VCs. Building on both empirical and theoretical insights, we propose NeuroQuant, a state-of-the-art PTQ approach for INR-VC that enables variable-rate coding without complex re-training. Our contributions tackle key challenges through the following research questions:

1. **How to realize variable bitrate** (Sec. 3.1): We redefine variable-rate coding as a mixed-precision quantization problem. By theoretically demonstrating that the assumption of inter-layer independence (Dong et al., 2020; Guan et al., 2024) does not apply to non-generalized models, we highlight the necessity of incorporating weight perturbation directionality and off-diagonal Hessian information for sensitivity assessment in quantizing INR-VC. Additionally, we introduce the Hessian-Vector product to simplify computations by eliminating the need for explicit Hessian calculations.

2. **How to ensure reconstruction quality** (Sec. 3.2): We enhance reconstruction quality by calibrating the QPs on the corresponding video-specific weights. Through second-order analysis, we derive a unified formula for MSE-oriented calibration across varying granularities. By considering significant cross-layer dependencies and the diverse distribution of weights, we conduct network-wise calibration and channel-wise quantization to minimize reconstruction loss.

3. **How NeuroQuant performs** (Sec. 4): We benchmark proposed NeuroQuant across various architectures against existing quantization techniques, achieving state-of-the-art results. For variable-rate coding, NeuroQuant outperforms competitors while reducing encoding time by 80%. Moreover, NeuroQuant is able to quantize weights down to INT2 without notable performance degradation.

4. **How to advance INR-VC** (Sec. 3.3): We revisit INR-VC through the lens of variational inference, proposing that the success of NeuroQuant stems from resolving the mismatch between the representation and compression. We also suggest that rate-distortion (R-D) optimization is also applicable to INR-VC and has the potential to achieve improved performance.

---

[1]To avoid ambiguity, we use the term *calibration* to describe the process of optimizing QPs, though some literature refers to this as *reconstruction*. In this paper, $reconstruction$ refers to the decoded video from INR-VC system. And for simplicity, layer calibration also stands for block calibration.

## 2 PRELIMINARIES

**Basic Notations.** We follow popular notations used in neural network. Vectors are denoted by lowercase bold letters, while matrices (or tensors) are denoted by uppercase bold letters. For instance, $\boldsymbol{W}$ refers to the weight tensor, and $\boldsymbol{w}$ is its flattened version. The superscript of $\boldsymbol{w}^{(l)}$ indicates the layer index. For a convolutional or a fully-connected layer, we mark input and output vectors by $\boldsymbol{x}$ and $\boldsymbol{z}$. Given a feedforward neural network with $n$ layers, the forward process is expressed as

$$\boldsymbol{x}^{(l+1)} = h(\boldsymbol{z}^{(l)}) = h(\boldsymbol{w}^{(l)}\boldsymbol{x}^{(l)}),\ 1 \leq l \leq n, \tag{1}$$

where $h(\cdot)$ denotes the activation function. For simplicity, we omit the additive bias, merging it into the activation. In the following, the notation $||\cdot||$ represents the Frobenius norm. Suppose $\boldsymbol{x}$ is sampled from the dataset $\mathcal{X}$, then the overall task loss is expressed as $\mathbb{E}_{\boldsymbol{x}\sim\mathcal{X}}[\mathcal{L}(\boldsymbol{w},\boldsymbol{x})]$.

**INR-based Video Coding.** INR-VC operates on the principle that a target video can be encoded into learned weights through end-to-end training. For each frame $V_t$ in an RGB video sequence $\mathbb{V} = \{V_t\}_{t=1}^{T} \in \mathbb{R}^{T \times 3 \times H \times W}$, INR-VC assumes the existence of an implicit continuous mapping $\mathcal{F} : [0,1]^{d_{in}} \rightarrow \mathbb{R}^{d_{out}}$ in the real-world system such that $V_t = \mathcal{F} \circ t$. According to the Universal Approximation Theorem (Hanin, 2019; Park et al., 2021), the unknown $\mathcal{F}$ can be approximated by a neural network $\mathcal{D}$ of finite length $L_\mathcal{D}$. The estimated $\hat{V}_t$ is then expressed as:

$$\hat{V}_t = \mathcal{D} \circ \mathcal{E}(t) = U_L \circ h \circ U_{L-1} \circ \cdots \circ h \circ U_1 \circ \mathcal{E}(t), \tag{2}$$

where $\mathcal{D}$ consists of cascaded upsampling layers $U$, and $\mathcal{E}(\cdot)$ is an embedding of the timestamp $t$. Typically, index-based INR-VCs (Chen et al., 2021a) employ a fixed Positional Encoding function or a learnable grid (Lee et al., 2023) as $\mathcal{E}(\cdot)$, while content-based INR-VCs (Chen et al., 2023; Zhao et al., 2023) utilize a learnable encoder. The encoding of INR-VC involves training the learnable weights $\boldsymbol{w}$ and subsequently compressing $\boldsymbol{w}$ into a bitstream using quantization and entropy coding techniques. While existing INR-VC works primarily focus on minimizing distortion during the training stage, video coding is fundamentally a R-D trade-off.

**Post-Training Quantization.** PTQ offers a push-button solution to quantize pretrained models without weights training. It contrasts with Quantization-Aware Training (QAT), which involves both weight optimization and quantization during training, leading to huge training cost. PTQ is generally a two-step process: 1) initializing QPs (e.g., steps) with allocated bitwidth and weight distribution statistics; 2) calibrating QPs to reduce quantization-induced loss. PTQ typically employs uniform affine transformation to map continuous $w \in \mathbb{R}$ to fixed-point integers $\hat{w}$. Traditional methods aim to minimize quantization error $||\hat{w} - w||$. However, an increasing number of explorations (Stock et al., 2020; Nagel et al., 2020; Hubara et al., 2021) suggest that this approach can yield sub-optimal results, as the parameter space error does not equivalently reflect task loss. To analyze quantization-induced loss degradation, AdaRound (Nagel et al., 2020) interprets quantization error as weight perturbation, i.e., $\hat{\boldsymbol{w}} = \boldsymbol{w} + \Delta\boldsymbol{w}$. The loss degradation can be approximated using Taylor series:

$$\mathbb{E}[\mathcal{L}(\boldsymbol{w} + \Delta\boldsymbol{w}, \boldsymbol{x}) - \mathcal{L}(\boldsymbol{w}, \boldsymbol{x})] \approx \Delta\boldsymbol{w}^T \cdot \boldsymbol{g}^{(\boldsymbol{w})} + \frac{1}{2}\Delta\boldsymbol{w}^T \cdot \boldsymbol{H}^{(\boldsymbol{w})} \cdot \Delta\boldsymbol{w}, \tag{3}$$

where $\boldsymbol{g}^{(\boldsymbol{w})} = \mathbb{E}[\nabla_{\boldsymbol{w}}\mathcal{L}]$ and $\boldsymbol{H}^{(\boldsymbol{w})} = \mathbb{E}[\nabla_{\boldsymbol{w}}^2\mathcal{L}]$ represent expected gradient and the second-order Hessian matrix, respectively. For well-converged weights, gradients tend to be close to $0$. AdaRound further assumes inter-layer independence, leading to a diagonal Hessian matrix optimization. BRECQ (Li et al., 2021b) extends AdaRound's layer-wise calibration to block granularity based on inter-block independence. However, these methods can significantly degrade the performance of non-generalized INR-VCs, which exhibit significant dependencies among layers.

**Mixed-Precision Quantization.** Mixed-precision quantization facilitates fine-grained rate control in INR-VCs, with bit allocation being crucial due to the varying levels of redundancy across layers and their different contributions to overall performance. However, determining optimal bitwidth assignments presents a significant challenge because of the extensive search space. For a network with $N$ layers and $M$ candidate bitwidths per layer, exhaustive combinatorial searches exhibit exponential time complexity of $\mathcal{O}(M^N)$. To address it, various strategies have been explored, including search-based reinforcement learning Wang et al. (2019); Lou et al. (2019), neural architecture search (Wu et al., 2016), and Hessian-based criteria (Dong et al., 2019; 2020). Despite these efforts, they often prove impractical for INR-VCs, as the search costs may surpass those of retraining a model. Furthermore, many existing criteria lack a robust theoretical basis for their optimality, rendering them less reliable in INR-VC systems.

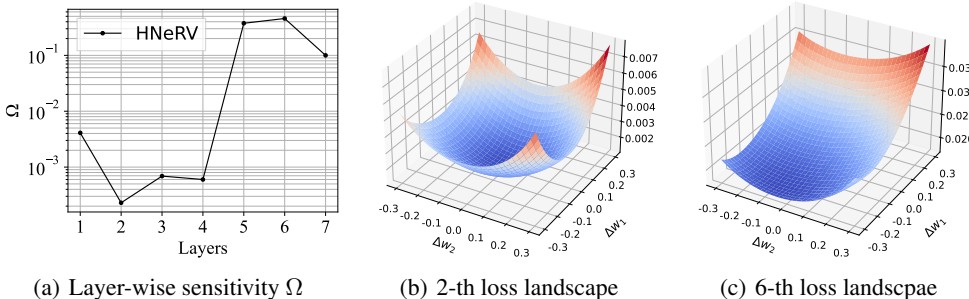

(a) Layer-wise sensitivity $\Omega$      (b) 2-th loss landscape      (c) 6-th loss landscpae

Figure 2: Examples of quantizing layers in sequence. (a) Different layers exhibits varying sensitivities. (b) Lower $\Omega$ means flatter loss landscape. (c) Higher $\Omega$ is otherwise, and the loss landscape shows pronounced directivity, indicating the necessity of considering the direction of $\Delta \boldsymbol{w}$.

## 3 METHODOLOGY

We introduce the proposed NeuroQuant for high-performance variable-rate INR-VC as follows:

**Problem 1** (NeuroQuant). *Given learned video-specific weights, the objective of NeuroQuant is to achieve different R-D trade-offs by quantizing post-training weights with variable QPs. This can be formulated as a rate-constrained optimization process:*

$$\arg\min \mathbb{E}[\mathcal{L}(\mathcal{Q}(\boldsymbol{w}), \mathcal{Q}(\boldsymbol{e})) - \mathcal{L}(\boldsymbol{w}, \boldsymbol{e})] \tag{4}$$

$$s.t. \ \sum_{l=1}^{L} Param(\boldsymbol{w}^{(l)}) \cdot b_{\boldsymbol{w}}^{(l)} + \sum_{t=1}^{T} Param(\boldsymbol{e}^{(t)}) \cdot b_{\boldsymbol{e}} = \mathcal{R} \pm \epsilon, \tag{5}$$

*where $\mathcal{R}$ represents the target bitrate, $\boldsymbol{e}$ denotes the embedding, $Param(\cdot)$ indicates the number of parameters, and $b$ denotes the bitwidth.*

We decouple this problem into three sub-problems: 1)Sec. 3.1: The rate-constrained term in Eq. 5 is defined as a mixed-precision bit assignment problem, accounting for fine-grained rate control and varying layer sensitivity; 2) Sec. 3.2: The objective in Eq. 4 is interpreted as QP calibration problem, focusing on calibration and quantization granularity of non-generalized INR-VC; 3) Sec. 3.3: We revisit the entire problem from the perspective of variational inference to provide a broader theoretical grounding.

### 3.1 HOW TO REALIZE VARIABLE BITRATE

**Sensitivity Criterion.** The core concept of mixed-precision quantization is to allocate higher precision (e.g., greater bitwidth) to sensitive layers while reducing precision in insensitive ones. Sensitivity can be intuitively understood through the flatness of the loss landscape (Li et al., 2018), as illustrated in Fig. 2. A flatter landscape, indicating lower sensitivity, corresponds to smaller loss changes with weight perturbations, whereas a sharper landscape indicates otherwise. Sensitivity essentially captures the curvature of the loss function, often described using second-order information, particularly the Hessian matrix $\boldsymbol{H}^{(w)}$. $\boldsymbol{H}^{(\boldsymbol{w})}$ defines how perturbations in weights affect task loss. For instance, HAWQ (Dong et al., 2019) uses the top Hessian eigenvalue as a sensitivity criterion, while HAWQ-V2 (Dong et al., 2020) demonstrates that the trace offers a better measure. However, these criteria rely on two key assumptions: 1) **Layer Independence**: Layers are mutually independent, allowing $\boldsymbol{H}^{(w)}$ to be treated as diagonal. 2) **Isotropy**: The loss function is directionally uniform under weight perturbations $\Delta \boldsymbol{w}$, meaning only $\boldsymbol{H}^{(w)}$ is considered, ignoring $\Delta \boldsymbol{w}$.

While these assumptions may hold for general-purpose networks, they break down in the context of non-generalized INR-VC, where significant inter-layer dependencies (Fig. 3(c)) and anisotropic behavior (Fig. 2(c)) exist. The following toy examples demonstrate why relying solely on diagonal information from $\boldsymbol{H}$ is suboptimal.

**Example 1** (Inter-Layer Dependencies). *Consider three functions, $\mathcal{F}_1 = 4x^2 + y^2$, $\mathcal{F}_2 = 4x^2 + 2y^2$, and $\mathcal{F}_3 = 4x^2 + 2y^2 + 5xy$. Their corresponding Hessians are given as:*

$$\boldsymbol{H}^{(\mathcal{F}_1)} = \begin{bmatrix} 8 & 0 \\ 0 & 2 \end{bmatrix}, \quad \boldsymbol{H}^{(\mathcal{F}_2)} = \begin{bmatrix} 8 & 0 \\ 0 & 4 \end{bmatrix}, \quad \boldsymbol{H}^{(\mathcal{F}_3)} = \begin{bmatrix} 8 & 5 \\ 5 & 4 \end{bmatrix}. \tag{6}$$

*All three functions share the same top eigenvalue (8), yet $\mathcal{F}_2$ and $\mathcal{F}_3$ are clearly more sensitive than $\mathcal{F}_1$. Although $\mathcal{F}_2$ and $\mathcal{F}_3$ have the same trace (12), $\mathcal{F}_3$ exhibits greater sensitivity due to the presence of off-diagonal terms (i.e., $5xy$).*

This demonstrates that inter-layer dependencies are overlooked when relying solely on diagonal information (e.g., eigenvalues or traces). Off-diagonal terms are essential to accurately capture sensitivity, highlighting the need to consider the full Hessian matrix. The story does not end there.

**Example 2** (Weight Perturbation Directions). *Assuming a perturbation $[\Delta x, \Delta y]$ applied to $\mathcal{H}^{(\mathcal{F}_3)}$ from above, the increase in loss is approximately proportional to*

$$\mathcal{F}_3(x + \Delta x, y + \Delta y) - \mathcal{F}_3(x, y) \approx [\Delta x, \Delta y]\boldsymbol{H}[\Delta x, \Delta y]^T = 8\Delta x^2 + 4\Delta y^2 + 10\Delta x \Delta y. \tag{7}$$

*Now, consider two cases: 1) Lower perturbation: $[\Delta x, \Delta y] = [0.1, 0.1]$; 2) Higher perturbation: $[\Delta x, \Delta y] = [0.2, -0.2]$. The increases in task loss are $0.22$ and $0.08$, respectively. Surprisingly, the higher perturbation results in a smaller task loss.*

This counterintuitive behavior is also observed in practice, where quantizing layers with higher $\boldsymbol{H}$ sensitivity to a lower bitwidth does not necessarily lead to significant performance degradation. We argue that allocating higher bitwidth to layers primarily reduces $||\Delta \boldsymbol{w}||$. However, this does not always guarantee a lower task loss, as $\mathcal{L}$ is anisotropy under $\Delta \boldsymbol{w}$ in INR-VC. The key insight is that task loss also depends on the direction of $\Delta \boldsymbol{w}$, not just its magnitude $||\Delta \boldsymbol{w}||$.

In conclusion, the sensitivity criterion of INR-VC must account for both the full Hessian matrix $\boldsymbol{H}^{(w)}$ and the direction of weight perturbations $\Delta \boldsymbol{w}$. This leads to the following theorem:

**Theorem 1.** *Assuming the INR-VC weights are twice differentiable and have converged to a local minima such that the first and second order optimality conditions are satisfied (i.e., the gradients are zero and the Hessian is positive semi-definite), the optimal sensitivity criteria for mixed-precision INR-VC is given by weighted Hessian information $\Omega = \Delta \boldsymbol{w}^T \cdot \boldsymbol{H}^{(w)} \cdot \Delta \boldsymbol{w}$.*

The criteria $\Omega$, formed by Hessian-Vector product, can essentially be interpreted as a linear transformation on $\boldsymbol{H}^{(w)}$, accounting for $\boldsymbol{H}^{(w)}$ along the weight perturbation directions. Existing Hessian-based criteria can be viewed as a degraded version of the proposed $\Omega$ that neglects the off-diagonal terms. For instance, Eq. 7 would degrade to $8\Delta x^2 + 4\Delta y^2$, and thus, loss is independent of inter-variable dependencies and perturbation direction.

**Approximating Hessian-Vector Product.** The Hessian matrix is challenging to explicitly compute and store as its quadratic complexity relative to the number of weights. Instead of forming $\boldsymbol{H}^{(w)}$ explicitly, we focus on the sensitivity criterion $\Omega = \Delta \boldsymbol{w}^T \cdot \boldsymbol{H}^{(w)} \cdot \Delta \boldsymbol{w}$. Let's construct a function of the form $\mathcal{G} = \boldsymbol{g}\Delta \boldsymbol{w}$, where $\boldsymbol{g}$ is the gradient of $\mathcal{L}$ with respect to $\boldsymbol{w}$. The gradient of $\mathcal{G}$ can be expressed as:

$$\nabla_{\boldsymbol{w}}\mathcal{G} = \frac{\partial \boldsymbol{g}\Delta \boldsymbol{w}}{\partial \boldsymbol{w}} = \frac{\partial \boldsymbol{g}}{\partial \boldsymbol{w}}\Delta \boldsymbol{w} + \boldsymbol{g}\frac{\partial \Delta \boldsymbol{w}}{\partial \boldsymbol{w}} = \frac{\partial^2 \mathcal{L}}{\partial \boldsymbol{w}^2}\Delta \boldsymbol{w} + \boldsymbol{g}\frac{\partial \Delta \boldsymbol{w}}{\partial \boldsymbol{w}} = \boldsymbol{H}^{(w)}\Delta \boldsymbol{w} + \boldsymbol{g}\frac{\partial \Delta \boldsymbol{w}}{\partial \boldsymbol{w}}. \tag{8}$$

In a converged model, $\boldsymbol{g}$ approaches 0. Moreover, quantization error can be modeled as a random vector, with its component sampled independently form a Uniform distribution: $\Delta \boldsymbol{w} \sim U(-0.5, 0.5)$ (Ballé et al., 2017). Thus, the second term in Eq. 8 can be ignored. This approximation is also akin to straight-through estimator (STE) (Liu et al., 2022), where $\frac{\partial \hat{w}}{\partial w} = \frac{\partial w}{\partial w}$ leads to $\frac{\partial \Delta \boldsymbol{w}}{\partial \boldsymbol{w}} = 0$. Consequently, we arrive at the final formulation for $\Omega$:

$$\Omega = \mathbb{E}[\Delta \boldsymbol{w}^T \nabla_{\boldsymbol{w}}\mathcal{G}], \quad where \quad \mathcal{G} = \boldsymbol{g}\Delta \boldsymbol{w} = \mathbb{E}[\nabla_{\boldsymbol{w}}\mathcal{L}\Delta \boldsymbol{w}], \, \mathcal{G} \in \mathbb{R}^1. \tag{9}$$

In Eq. 9, $\Delta \boldsymbol{w}$ is treated as a perturbation around $\boldsymbol{w}$, allowing us to compute $\boldsymbol{g}$ centered at $\boldsymbol{w}$. For each potential bitwidth configuration, we only need to compute $\Delta \boldsymbol{w}$ and the gradient of $\mathcal{G}$ in linear time. Notably, different from using $\mathcal{L}$ directly, such criteria-based methods do not require supervised labels or forward inference over the entire full datasets for each potential bitwidth candidate, enabling efficient mixed-precision search using techniques like integer programming, genetic algorithms (Guo et al., 2020), or iterative approaches. So far, we have realized bit allocation for a target bitrate. The next step involves calibrating QPs to minimize the reconstruction distortion.

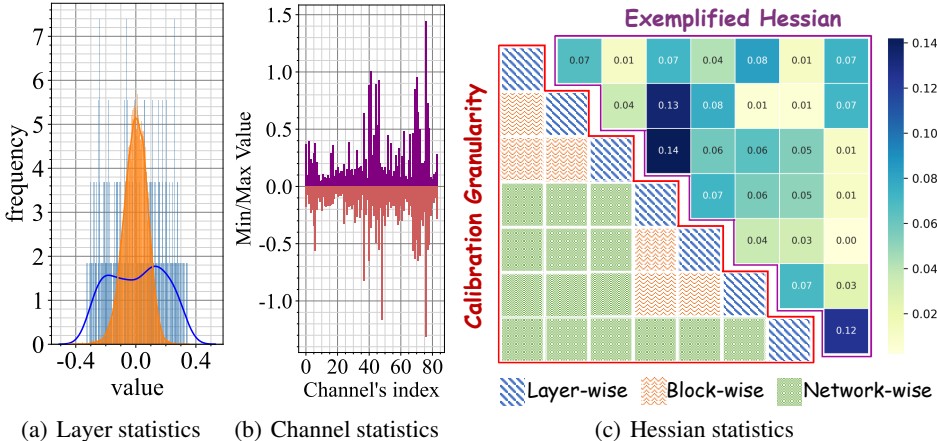

(a) Layer statistics    (b) Channel statistics    (c) Hessian statistics

Figure 3: Statistic of the weight distribution among (a) layers and (b) channels. Other layers/channels exhibit similar distributions. (c) Non-generalized representation shows obvious layer/block dependencies, implying that layer/block-wise calibration is not suitable for INR-VC. Exemplified statistic information is based on HNeRV (1M) in Beauty sequence.

## 3.2 How to ensure reconstruction quality

We follow the principle of PTQ to calibrate QPs (including steps and rounding variables in Eq. 16) without optimizing the underlying weights. PTQ allows us to preserve reconstruction quality by only calibrating QPs, instead of engaging in complex weight training as QAT.

**Unified Calibration Objective.** We begin by investigating the unified calibration objective. The quantization-induced task loss degradation of a well-converged model can be approximated using the second-order Taylor expansion:

$$\mathbb{E}[\mathcal{L}(\boldsymbol{w} + \Delta\boldsymbol{w}, \boldsymbol{x}) - \mathcal{L}(\boldsymbol{w}, \boldsymbol{x})] \approx \frac{1}{2}\Delta\boldsymbol{w}^T \cdot \boldsymbol{H}^{(w)} \cdot \Delta\boldsymbol{w}. \tag{10}$$

From this estimation, we aim to calibrate QPs by minimizing the proxy loss, defined as: $\min \Delta\boldsymbol{w}^T \cdot \boldsymbol{H}^{(w)} \cdot \Delta\boldsymbol{w}$. Denote the neural network output as $\boldsymbol{z}^{(n)}$. Using the chain rule, we can compute the Hessian matrix $\boldsymbol{H}^{(w)}$ as follows:

$$\frac{\partial\mathcal{L}^2}{\partial\boldsymbol{w}_i\partial\boldsymbol{w}_j} = \frac{\partial}{\partial\boldsymbol{w}_j}\sum_{k=1}^{m}\frac{\partial\mathcal{L}}{\partial\boldsymbol{z}_k^{(n)}}\frac{\partial\boldsymbol{z}_k^{(n)}}{\partial\boldsymbol{w}_i} = \sum_{k=1}^{m}\frac{\partial\mathcal{L}}{\partial\boldsymbol{z}_k^{(n)}}\frac{\partial^2\boldsymbol{z}_k^{(n)}}{\partial\boldsymbol{w}_i\partial\boldsymbol{w}_j} + \sum_{k,l=1}^{m}\frac{\partial\boldsymbol{z}_k^{(n)}}{\partial\boldsymbol{w}_i}\frac{\partial^2\mathcal{L}}{\partial\boldsymbol{z}_k^{(n)}\partial\boldsymbol{z}_l^{(n)}}\frac{\partial\boldsymbol{z}_l^{(n)}}{\partial\boldsymbol{w}_j}. \tag{11}$$

Since the post-training model converges to a local minimum, we can assume the Hessian is positive-semi-definite (PSD). With $\nabla_{\boldsymbol{z}^{(n)}}\mathcal{L}$ close to 0, the first term in Eq. 11 is negligible (Martens, 2010), yielding Gaussian-Newton form $\boldsymbol{G}^{(w)}$:

$$\boldsymbol{H}^{(w)} \approx \boldsymbol{G}^{(w)} = \boldsymbol{J}_{\boldsymbol{z}(n)}^T \boldsymbol{H}^{(\boldsymbol{z}^{(n)})} \boldsymbol{J}_{\boldsymbol{z}(n)}, \tag{12}$$

where $\boldsymbol{J}_{\boldsymbol{z}(n)}$ is the Jacobian matrix of the network output $\boldsymbol{z}^{(n)}$ with respect to the weights $\boldsymbol{w}$.

Considering $\mathcal{L}$ is given by a commonly used mean squared error (MSE), we have:

$$\boldsymbol{H}^{(\boldsymbol{z}^{(n)})} = 2\boldsymbol{I}_m \quad s.t. \quad \mathcal{L} = \sum_{k=1}^{m}(\boldsymbol{z}_k^{(n)} - x_k)^2. \tag{13}$$

By substituting the Eq. 12 and Eq. 13 into the Eq. 10, we get the following minimization objective:

$$\min \Delta\boldsymbol{w}^T \cdot \boldsymbol{H}^{(w)} \cdot \Delta\boldsymbol{w} \approx \min[\boldsymbol{J}_{\boldsymbol{z}(n)}\Delta\boldsymbol{w}]^T[\boldsymbol{J}_{\boldsymbol{z}(n)}\Delta\boldsymbol{w}] \approx \min \mathbb{E}[||\Delta\boldsymbol{z}^{(n)}||^2]. \tag{14}$$

**Remark 1.** *This results in a unified form of MSE-oriented calibration. Other distortion metrics may have analogous form, but this is not guaranteed, as not all metrics necessarily correspond to an analytical Hessian form. This can be addressed through approximated Hessian (Appendix C).*

**Calibration Granularity.** Based on the above objctive, we further investigate the optimal granularity for calibration. Following Li et al. (2021b); Wei et al. (2022), we can define:

1. **Layer-wise Calibration:** This assumes layers are mutually independent and calibrates each layers individually, resulting in a layer-diagonal Hessian matrix (blue diagonal in Fig. 3(c)), where $\Delta z$ in Eq. 14 reflects to the output error of individual layers.

2. **Block-wise Calibration:** A block is defined as a collection of several layers (e.g., residual blocks). This calibration considers intra-block dependencies while assuming inter-block independence. Calibration is performed block by block, leading to a block-diagonal Hessian matrix (orange diagonal in Fig. 3(c)), with $\Delta z$ representing the output error of the block as a whole.

3. **Network-wise Calibration:** This granularity calibrates entire quantized network by considering the global Hessian (full matrix in Fig. 3(c)), where $\Delta z$ reflects the entire network's output error.

In generalized networks, layer/block-wise calibration is widely accepted, as dependencies are primarily found within layers/blocks, allowing inter-layer/block dependencies to be neglected (Li et al., 2021b). However, their network-wise calibration may lead to poor performance due to high generalization error. In the context of INR-VC, the situation differs: inter-layer/block dependencies cannot be ignored (Fig. 3(c)), and INR-VC typically prioritizes non-generalized representation over generalization. Therefore, we advocate for network-wise calibration as our preferred approach, as it better captures the dependencies across the network layers.

**Quantization Granularity.** Next, we consider the granularity for quantization steps. Weight distributions vary significantly across different layers of a given INR-VC weights (Fig. 3(a)), and even among channels within a specific layer (Fig. 3(b)). This variability suggests that weights should be modeled on a channel-wise basis, where all weights in a channel share the same quantization step.

**Calibration.** Once the granularity of both calibration and quantization are determined, we next calibrate QPs to minimize distortion. While solving Eq. 14 circumvents the complexity issues associated with the Hessian, it remains an discrete optimization problem. Inspired by Nagel et al. (2020), we reformulate it into a continuous optimization framework using soft weight variables:

$$\arg \min_{s,v} \underbrace{||\mathcal{F}(\boldsymbol{x},\boldsymbol{w}) - \mathcal{F}(\boldsymbol{x},\tilde{\boldsymbol{w}})||^2}_{\text{Distortion term}\ \ \mathcal{L}_D = ||\Delta\boldsymbol{z}^{(n)}||^2} + \lambda \underbrace{\sum 1 - |2h(\boldsymbol{v}_i) - 1|^\beta}_{\text{Regularization term}\ \ \mathcal{L}_{Reg}}, \tag{15}$$

$$s.t. \quad \tilde{\boldsymbol{w}} = \boldsymbol{s} \cdot clip\left(\left\lfloor \frac{\boldsymbol{w}}{\boldsymbol{s}} \right\rfloor + h(\boldsymbol{v}), \ -2^{b-1}, \ 2^{b-1} - 1\right), \tag{16}$$

where $\boldsymbol{V}_{i,j}$ represents the continuous variable to optimize, and $h(\boldsymbol{V}_{i,j})$ is any differentiable function constrained between 0 and 1, i.e., $h(\boldsymbol{V}_{i,g}) \in [0,1]$. $\mathcal{L}_{Reg}$ acts as a differentiable regularizer, guiding $h(\boldsymbol{V}_{i,j})$ to converge towards either 0 or 1, ensuring at convergence $h(\boldsymbol{V}_{i,j}) \in \{0,1\}$. We also anneal $\beta$ in $\mathcal{L}_{reg}$ to facilitate stable convergence. This approach yields an intriguing observation: by minimizing the discrepancy between a high-precision teacher model and an initialized student model, we can effectively calibrate the quantized network. However, it's crucial to emphasize that we are different from knowledge distillation (Polino et al., 2018) that requires similar computational and data resources as naive training, making it impractical for our variable-rate coding.

### 3.3 HOW TO ADVANCE INR-VC

Our NeuroQuant adjusts QP to manage R-D trade-off where higher rates lead to lower distortion and vice versa. This trade-off can be formally interpreted through variational inference (Kingma, 2013; Ballé et al., 2017), aiming to approximate the true posterior $p_{\tilde{\boldsymbol{w}}|\boldsymbol{x}}(\tilde{\boldsymbol{w}}|\boldsymbol{x})$ with a variational density $q(\tilde{\boldsymbol{w}}|\boldsymbol{x})$ by minimizing the Kullback–Leibler (KL) divergence over the data distribution $p_{\boldsymbol{x}}$:

$$\mathbb{E}_{\boldsymbol{x}} D_{KL}[q||p_{\tilde{\boldsymbol{w}}|\boldsymbol{x}}] = \mathbb{E}_{\boldsymbol{x}\sim p_{\boldsymbol{x}}} \mathbb{E}_{\tilde{\boldsymbol{w}}\sim q}[\underbrace{\log q(\tilde{\boldsymbol{w}}|\boldsymbol{x})}_{=0} - \underbrace{\log p_{\boldsymbol{x}|\tilde{\boldsymbol{w}}}(\boldsymbol{x}|\tilde{\boldsymbol{w}})}_{\text{Distortion}\ \mathcal{L}_D} - \underbrace{\log p_{\tilde{\boldsymbol{w}}}(\tilde{\boldsymbol{w}})}_{\text{Rate}\ \mathcal{L}_R} + \underbrace{\log p_{\boldsymbol{x}}(\boldsymbol{x})}_{\text{const}}]. \tag{17}$$

This leads to the R-D trade-off for INR-VC (Appendix B):

$$\mathcal{L} = \mathcal{L}_R + \lambda\mathcal{L}_D = \log p_{\tilde{\boldsymbol{w}}}(\tilde{\boldsymbol{w}}) + \frac{1}{2\sigma^2}||\boldsymbol{x} - \hat{\boldsymbol{x}}||^2, \ \ s.t. \ p_{\boldsymbol{x}|\tilde{\boldsymbol{w}}}(\boldsymbol{x}|\tilde{\boldsymbol{w}}) = \mathcal{N}(\boldsymbol{x}|\tilde{\boldsymbol{x}}, \sigma^2), \tag{18}$$

In coding, $\tilde{\boldsymbol{w}}$ is replaced by the discrete symbol $\hat{\boldsymbol{w}}$, which can be losslessly compressed using entropy coding techniques, such as arithmetic coding (Rissanen & Langdon, 1981).

**Remark 2** (NeuroQuant vs. INR-VC). *Popular INR-VCs (Chen et al., 2021a; Li et al., 2022b; He et al., 2023; Zhao et al., 2023; 2024) focus on* $\min \log p_{\boldsymbol{x}|\boldsymbol{w}}(\boldsymbol{x}|\boldsymbol{w})$ *without considering the impact of weight quantization, creating a mismatch between representation and compression objectives. i.e.,* $p(\boldsymbol{x}|\boldsymbol{w})$ *and* $p(\boldsymbol{x}|\tilde{\boldsymbol{w}})$, *which degrades performance after quantization. In contrast, the proposed NeuroQuant directly optimizes* $\log p_{\boldsymbol{x}|\tilde{\boldsymbol{w}}}(\boldsymbol{x}|\tilde{\boldsymbol{w}})$, *bridging this mismatch and yielding superior results.*

While QAT (Ladune et al., 2023; Kim et al., 2024) can also optimize the same objective, NeuroQuant's strength lies in transforming R-D optimization into a post-training process, particularly advantageous for efficient variable-rate video coding, thereby reducing encoding costs. Despite NeuroQuant enables a more flexible R-D trade-off than existing approaches, it is not yet a fully joint optimization framework. Recent works (Gomes et al., 2023; Zhang et al., 2024) have begun to recognize the necessity of joint R-D optimization in INR-VC, but they lack principled explanations and typically underperform compared to generalized codecs. For example, they often assume a Gaussian prior or i.i.d. context model (Minnen et al., 2018; He et al., 2021), which is not entirely correct for INR-VC. Future advancements will require the development of customized context models to better capture weight characteristics, enabling true joint R-D optimization for INR-VC systems.

# 4 EXPERIMENTS

In this section, we conduct thorough experiments to verify the effectiveness of NeuroQuant. Detailed implementation and additional experiments are available in Appendix E.

**INR-VC Baselines.** We select three representative INR-VCs as baselines to evaluate various quantization methods. NeRV (Li et al., 2022b) is a pioneering model that first maps frame indices to video frames. HNeRV (Chen et al., 2023) first introduces an encoder to generate learnable embeddings instead of the positional encoding used in NeRV. HiNeRV (Kwan et al., 2024a) further optimizes the network architecture, achieving start-of-the-art performance.

**Quantization Benchmarks.** We first introduce naive PTQ in HNeRV, a widely adopted approach across various INR-VCs. We also include four leading task-oriented PTQs: AdaRound (Nagel et al., 2020), BRECQ (Li et al., 2021b), QDrop (Wei et al., 2022), as well as RDO-PTQ (Shi et al., 2023), which is specifically designed for R-D optimization. Two QATs tailored for INR-VCs derived from FFNeRV (Lee et al., 2023) and HiNeRV (Kwan et al., 2024a) are also evaluated.

## 4.1 QUANTIZATION

We summarize the quantization performance in Table 1. Starting with direct quantization in HNeRV, the results confirm Remark 2: optimizing reconstructive representation without considering weight quantization (through training awareness or post-calibration) significantly degrades performance. When compared with leading task-oriented PTQs—AdaRound, BRECQ, QDrop, and RDOPTQ—NeuroQuant consistently demonstrates superior performance, with the advantage growing as bitwidth decreases. The results support our analysis: non-generalized INR-VC exhibits significant inter-layer dependencies, making network-wise calibration necessary. Our NeuroQuant also outperforms QATs designed for INR-VC (i.e., FFNeRV and HiNeRV) across all listed baselines, particularly at lower bitwidths, achieving gains of more than 3dB. Its adaptability to varying precision highlights superior bitrate flexibility, indicating its potential for variable-rate video coding.

## 4.2 VARIABLE-RATE CODING PERFORMANCE

Figure 4 depicts the R-D curves for the evaluated methods. Compared to NeRV and HNeRV equipped with direct 8-bit quantization, NeuroQuant demonstrates significant compression efficiency gains of 27.8% and 25.5%, respectively. This improvement primarily stems from their lack of optimization for the compression objective. In contrast, HiNeRV achieves superior compression performance by incorporating QAT, as discussed in Sec. 3.3. Despite this, NeuroQuant also brings another 4.8% gains by replacing the built-in QAT. A key advantage of NeuroQuant is its ability to support variable-rate coding without training separate weights for each target bitrate, offering both flexibility and efficiency in practice. Moreover, NeuroQuant outperforms the network coding tool DeepCABAC (Wiedemann et al., 2020), highlighting the benefits of task-oriented QPs. Incorporating its advanced entropy coding with NeuroQuant is an exciting avenue for future research.

Table 1: Reconstruction quality comparison of different quantization methods (vertical) across various INR-VCs with different model sizes (horizontal) in terms of PSNR on UVG (Mercat et al., 2020). Bold values indicates the best results. † denotes QAT strategies and * represents mixed precision. All implementations are based on open-source codes.

| Methods | W-bits | NeRV | NeRV | HNeRV | HNeRV | HiNeRV | Avg |
|---|---|---|---|---|---|---|---|
| Full Prec. (dB) | 32 | 31.39 | 32.30 | 32.49 | 33.80 | 35.09 | 33.01 |
| Param. | - | 3.1M | 6.2M | 3.0M | 6.2M | 3.1M | - |
| HNeRV (Chen et al., 2023) | 6 | 30.68 | 31.56 | 32.05 | 33.29 | 32.10 | 31.94 |
| FFNeRV (Lee et al., 2023)† | 6 | 31.10 | 32.02 | 32.15 | 33.34 | 34.03 | 32.53 |
| HiNeRV (Kwan et al., 2024a)† | 6 | 31.20 | 32.09 | 32.25 | 33.48 | 34.54 | 32.71 |
| AdaRound (Nagel et al., 2020) | 6 | 31.03 | 31.96 | 32.10 | 33.26 | 33.92 | 32.45 |
| BRECQ (Li et al., 2021b) | 6 | 31.11 | 32.05 | 32.18 | 33.42 | 34.10 | 32.57 |
| QDrop (Wei et al., 2022) | 6 | 31.15 | 32.10 | 32.20 | 33.44 | 34.27 | 32.63 |
| RDOPTQ (Shi et al., 2023) | 6 | 31.15 | 32.06 | 32.16 | 33.39 | 34.23 | 32.60 |
| NeuroQuant (Ours) | 6 | **31.31** | **32.22** | **32.38** | **33.61** | **34.67** | **32.84** |
| HNeRV (Chen et al., 2023) | 4 | 27.02 | 27.86 | 28.14 | 28.60 | 24.30 | 27.18 |
| FFNeRV (Lee et al., 2023)† | 4 | 30.14 | 30.90 | 31.11 | 32.13 | 32.37 | 31.33 |
| HiNeRV (Kwan et al., 2024a)† | 4 | 30.37 | 31.32 | 31.46 | 32.67 | 32.95 | 31.75 |
| AdaRound (Nagel et al., 2020) | 4 | 30.12 | 30.65 | 31.02 | 31.98 | 32.10 | 31.17 |
| BRECQ (Li et al., 2021b) | 4 | 30.22 | 30.93 | 31.26 | 32.24 | 32.56 | 31.44 |
| QDrop (Wei et al., 2022) | 4 | 30.28 | 31.05 | 31.33 | 32.45 | 32.68 | 31.56 |
| RDOPTQ (Shi et al., 2023) | 4 | 30.25 | 30.96 | 31.24 | 32.25 | 32.60 | 31.46 |
| NeuroQuant (Ours) | 4 | **30.85** | **31.77** | **31.64** | **32.81** | **33.33** | **32.08** |
| HNeRV (Chen et al., 2023) | 2 | 14.81 | 15.50 | 13.32 | 13.06 | 13.30 | 14.00 |
| FFNeRV (Lee et al., 2023)† | 2 | 22.52 | 22.89 | 23.84 | 24.14 | 21.25 | 22.93 |
| HiNeRV (Kwan et al., 2024a)† | 2 | 24.08 | 25.30 | 25.11 | 26.51 | 23.89 | 24.98 |
| AdaRound (Nagel et al., 2020) | 2 | 22.51 | 23.44 | 23.70 | 24.56 | 22.10 | 23.26 |
| BRECQ (Li et al., 2021b)* | 2 | 24.05 | 25.17 | 25.40 | 26.32 | 25.87 | 25.36 |
| QDrop (Wei et al., 2022)* | 2 | 25.32 | 26.14 | 25.94 | 26.85 | 26.60 | 26.17 |
| RDOPTQ (Shi et al., 2023)* | 2 | 24.33 | 25.75 | 25.57 | 26.50 | 26.14 | 25.66 |
| NeuroQuant (Ours)* | 2 | **27.39** | **28.48** | **28.02** | **29.05** | **28.92** | **28.37** |

## 4.3 Diving into NeuroQuant

In this subsection, we perform a series of evaluations to gain deeper insights into how our NeuroQuant functions.

**Encoding Complexity.** Table 2 presents the encoding time evaluation for $960 \times 1920$ videos with approximate 3M parameters. From NeRV to HiNeRV, compression performance is improved at the cost of increased encoding complexity. Without support for variable bitrates, generating a new bitrate point typically requires retraining the model, leading to an encoding time that is generally unacceptable (e.g., exceeding 22 hours for HiNeRV). In this context, NeuroQuant provides a practical solution by leveraging a pre-trained model, enabling adaptation to variable bitrates while ensuring efficient encoding, achieving speedups of up to 7.9 times.

Furthermore, we select the Jockey sequence from UVG as a representative example to have an analysis of variable-rate coding in Fig 5. The left subplot presents the available points of mixed precision versus unified precision. The middle sub-

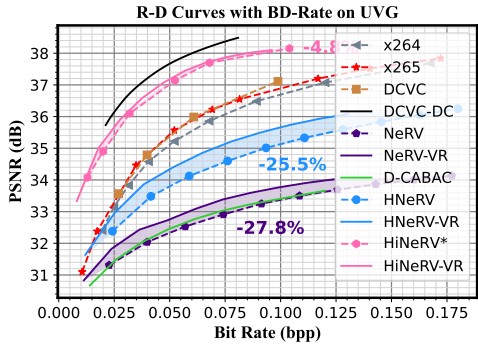

Figure 4: Compression efficiency comparison. Variable-rate models are labeled with -*VR* suffix using the solid line. * is INR-VC using QAT.

Table 2: Encoding time required to support a new bitrate. Note that our pretrained model is shared for all bitrates in range.

| Baselines | Naive | NeuroQuant | Pretrain |
|---|---|---|---|
| NeRV | 1.8 h | 0.4 h (×4.5) | 1.8 h |
| HNeRV | 4.7 h | 1.0 h (×4.7) | 4.7 h |
| HiNeRV | 22.2 h | 2.8 h (×7.9) | 18.9 h |

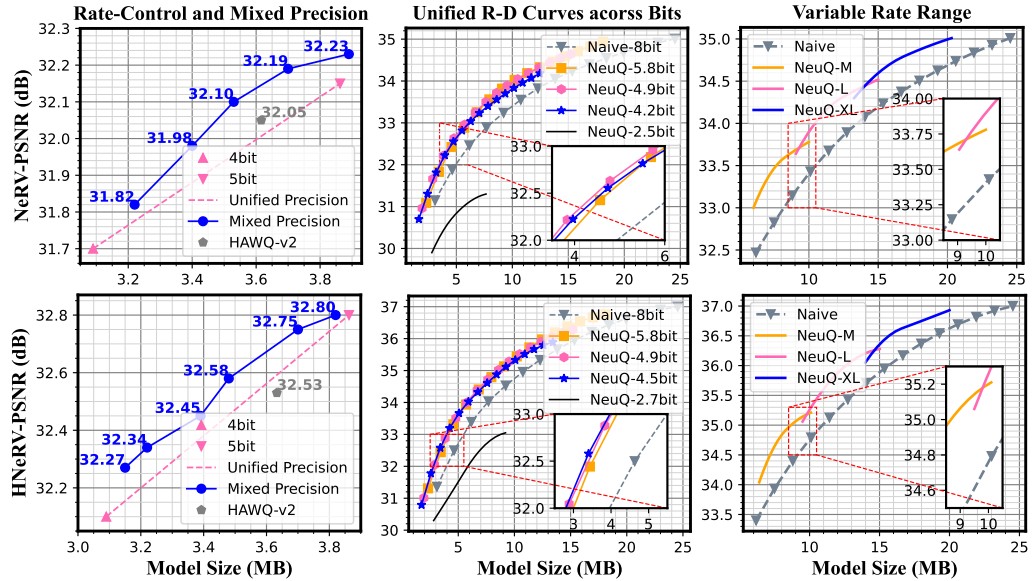

Figure 5: Quantitative comparison of NeRV (up) and HNeRV (bottom).

plot depicts the RD curves for different quantization bitwidths, while the right subplot displays the variable-rate range for models of varying sizes. More results can be found in Appendix E.3. We have the following conclusions:

**Rate Control and Mixed Precision.** Mixed precision supports finer-grained rate control compared to unified precision, making NeuroQuant more flexible for various bitrate. Additionally, it enables better bit allocation, resulting in improved coding efficiency.

**Unified R-D Curves across Bits.** The coarseness of quantization affects both rate and distortion, leading to a R-D trade-off. Within a certain range, different bitwidths exhibit unified R-D characteristics, providing a foundation for variable-rate coding through adjusting quantization parameters. While we demonstrate state-of-the-art performance, the degradation is still noticeable below INT3. As a result, we do not recommend INT3 and below for variable-rate scenarios.

**Variable-Rate Range.** While NeuroQuant significantly reduces the need for exhaustive individual weights training, achieving infinite rate range through quantization alone is impossible due to the inherent lower bound of bitwidth. We believe this limitation can be addressed by introducing pruning, which can be interpreted as a special case of NeuroQuant with $bitwidth = 0$. This approach opens up new possibilities for further exploration based on the foundations laid by this work.

## 5 CONCLUSIONS

**Conclusion.** In this work, we introduce NeuroQuant, a novel approach for variable-rate INR-VC that adjusts the QPs of post-training weights to control bitrate efficiently. Our key insight is that non-generalized INR-VC exhibits distinct characteristics for quantization. Through empirical and theoretical analysis, we establish the state-of-the-art weight quantization for INR-VC. NeuroQuant significantly reduces encoding complexity while maintaining leading compression performance, providing an effective solution for variable-rate video coding in neural representation.

**Limitations and Future Work.** Despite the success of NeuroQuant, there are some limitations. The variable-rate range is inherently constrained by the available bitwidths. A promising solution is integrating weight pruning techniques to endow the model weights with greater variability. Additionally, as discussed in Sec. 3.3, NeuroQuant enables variable R-D trade-offs than existing methods, but it still fall short of achieving true joint R-D optimization. Future work will explore customized probabilistic modeling for INR-VC, aiming to enable a fully joint R-D optimization process.

ACKNOWLEDGMENTS

This work was supported in part by National Key Research and Development Project of China (Grant No. 2022YFF0902402), Natural Science Foundation of Jiangsu Province (Grant No. BK20241226) and Natural Science Foundation of China (Grant No. 62401251, 62431011). The authors would like to express their sincere gratitude to the Interdisciplinary Research Center for Future Intelligent Chips (Chip-X) and Yachen Foundation for their invaluable support.

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

## A  VARIABLE-RATE BACKGROUND

In real-world communication scenarios, practical video codecs pursue variable-rate coding to adapt to complex network environments while reducing storage and computational overhead.

In traditional video codecs (Wiegand et al., 2003; Sullivan et al., 2012), variable-rate coding is typically achieved by adjusting quantization parameters (QPs) (Liu et al., 2008; Li et al., 2020). For learned generalized video codecs, current variable-rate methods can be categorized into two main approaches: feature modulation and multi-granularity quantization. Feature modulation focuses on modifying encoding and decoding features to achieve representations with varying entropy. For

instance, Lin et al. (2021) proposed a scaling network that modulates the internal feature maps of motion and residual encoders. Similarly, Rippel et al. (2021) converted discrete rate levels into one-hot vectors, which were then fed into various codec sub-modules. Duan et al. (2024) introduced an adaptive conditional convolution method that applies affine transformations to features based on input Lagrangian multipliers ($\lambda$). In contrast, multi-granularity quantization aims to control bitrate by adjusting the quantization levels of feature maps (also referred to as latents). For example, Li et al. (2022a) proposed a coarse-to-fine quantization strategy using three learnable quantization granularity parameters: global levels, channel-wise levels, and spatial-channel-wise levels.

However, INR-based video codecs (INR-VCs) take a fundamentally different approach, converting video signal compression into model weight compression. This diverges from the conventional focus on extracting compact feature representations, making tailored variable-rate coding strategies for INR-VCs essential. While preliminary studies have begun exploring variable-rate coding in INR-VCs, their performance remains limited (detailed in Sec. E.2).

In this paper, we reduce this gap by framing variable-rate coding for INR-VCs as a mixed-precision bit allocation problem. Our approach achieves variable-rate coding through QP adjustments and demonstrates state-of-the-art performance.

## B    VARIATIONAL INFERENCE PERSPECTIVE

In INR-VC, the variational inference framework interprets the rate-distortion (R-D) trade-off as an optimization problem, aiming to approximate the true posterior distribution $p_{\tilde{\boldsymbol{w}}|\boldsymbol{x}}(\tilde{\boldsymbol{w}}|\boldsymbol{x})$ with a variational density $q(\tilde{\boldsymbol{w}}|\boldsymbol{x})$ by minimizing the expected Kullback-Leibler (KL) divergence over the data distribution $p_{\boldsymbol{x}}$. Starting with the KL divergence, we have:

$$\mathbb{E}_{\boldsymbol{x}\sim p_{\boldsymbol{x}}}D_{KL}[q||p_{\tilde{\boldsymbol{w}}|\boldsymbol{x}}] = \mathbb{E}_{\boldsymbol{x}\sim p_{\boldsymbol{x}}}\mathbb{E}_{\tilde{\boldsymbol{w}}\sim q}\left[\log\frac{q(\tilde{\boldsymbol{w}}|\boldsymbol{x})}{p_{\tilde{\boldsymbol{w}}|\boldsymbol{x}}(\tilde{\boldsymbol{w}}|\boldsymbol{x})}\right] \tag{19}$$

$$= \mathbb{E}_{\boldsymbol{x}\sim p_{\boldsymbol{x}}}\mathbb{E}_{\tilde{\boldsymbol{w}}\sim q}\left[\log q(\tilde{\boldsymbol{w}}|\boldsymbol{x}) - \log p_{\tilde{\boldsymbol{w}}|\boldsymbol{x}}(\tilde{\boldsymbol{w}}|\boldsymbol{x})\right]. \tag{20}$$

Based on the Bayes' theorem, the posterior $p_{\tilde{\boldsymbol{w}}|\boldsymbol{x}}(\tilde{\boldsymbol{w}}|\boldsymbol{x})$ can be expressed as:

$$p_{\tilde{\boldsymbol{w}}|\boldsymbol{x}}(\tilde{\boldsymbol{w}}|\boldsymbol{x}) = \frac{p_{\boldsymbol{x}|\tilde{\boldsymbol{w}}}(\boldsymbol{x}|\tilde{\boldsymbol{w}})p_{\tilde{\boldsymbol{w}}}(\tilde{\boldsymbol{w}})}{p_{\boldsymbol{x}}(\boldsymbol{x})}. \tag{21}$$

Substituting this expression into Eq. 20, we get:

$$\mathbb{E}_{\boldsymbol{x}\sim p_{\boldsymbol{x}}}D_{KL}[q||p_{\tilde{\boldsymbol{w}}|\boldsymbol{x}}] = \mathbb{E}_{\boldsymbol{x}\sim p_{\boldsymbol{x}}}\mathbb{E}_{\tilde{\boldsymbol{w}}\sim q}[\log q(\tilde{\boldsymbol{w}}|\boldsymbol{x}) - \log p_{\boldsymbol{x}|\tilde{\boldsymbol{w}}}(\boldsymbol{x}|\tilde{\boldsymbol{w}}) - \log p_{\tilde{\boldsymbol{w}}}(\tilde{\boldsymbol{w}}) + \log p_{\boldsymbol{x}}(\boldsymbol{x})]. \tag{22}$$

Here, we arrive at a form consistent with Eq.17. Let's examine each of terms individually.

**First Term.** For unknown mapping from timestamps $\boldsymbol{t}$ to video data $\boldsymbol{x} = \mathcal{F}(\boldsymbol{w}, \boldsymbol{t})$, the *inference* refers to computing the inverse transformation from input $\boldsymbol{x}$ (Ballé et al., 2017). Using a soft variable similar to Eq. 16, we have:

$$q(\tilde{\boldsymbol{w}}|\boldsymbol{x}) = \prod_i \mathcal{U}(\tilde{\boldsymbol{w}}_i|\boldsymbol{w}_i - \frac{1}{2}, \boldsymbol{w}_i + \frac{1}{2}), \quad with \ \boldsymbol{w} = \mathcal{F}^{-1}\circ\boldsymbol{x}, \tag{23}$$

where $\mathcal{U}$ denotes a uniform distribution centered around $\boldsymbol{w}_i$. Then:

$$\mathbb{E}_{\tilde{\boldsymbol{w}}\sim q}[\log q(\tilde{\boldsymbol{w}}|\boldsymbol{x})] = \mathbb{E}_{\tilde{\boldsymbol{w}}\sim q}\left[\log\mathcal{U}\left(-\frac{1}{2}, \frac{1}{2}\right)\right] = 0. \tag{24}$$

**Second Term.** For the second term, suppose $p_{\boldsymbol{x}|\tilde{\boldsymbol{w}}}(\boldsymbol{x}|\tilde{\boldsymbol{w}})$ follows a normal distribution:

$$p_{\boldsymbol{x}|\tilde{\boldsymbol{w}}}(\boldsymbol{x}|\tilde{\boldsymbol{w}}) = \mathcal{N}(\boldsymbol{x}|\tilde{\boldsymbol{x}}, \sigma^2), \ \tilde{\boldsymbol{x}} = \mathcal{F}(\tilde{\boldsymbol{w}}, \boldsymbol{t}). \tag{25}$$

Maximizing $\log p_{\boldsymbol{x}|\tilde{\boldsymbol{w}}}(\boldsymbol{x}|\tilde{\boldsymbol{w}})$ is equivalent to minimizing the squared error term:

$$\max \log p_{\boldsymbol{x}|\tilde{\boldsymbol{w}}}(\boldsymbol{x}|\tilde{\boldsymbol{w}}) = -\min\log\mathcal{N}(\boldsymbol{x}|\tilde{\boldsymbol{x}}, \sigma^2) \tag{26}$$

$$= -\min\log\frac{1}{\sigma\sqrt{2\pi}}\exp\left(-\frac{1}{2\sigma^2}||\boldsymbol{x} - \tilde{\boldsymbol{x}}||^2\right) \tag{27}$$

$$= \min\frac{1}{2\sigma^2}||\boldsymbol{x} - \tilde{\boldsymbol{x}}||^2 \tag{28}$$

**Third Term.** The third term reflects the cost of encoding the latent variables $\tilde{w}$, representing the model complexity.

**Forth Term.** For a specific video data, the true distribution $p(x)$ is constant.

In conclusion, we obtain the objective function as derived in Eq. 17:

$$\mathcal{L} = \mathcal{L}_R + \lambda\mathcal{L}_D = \log p_{\tilde{w}}(\tilde{w}) + \frac{1}{2\sigma^2}||x - \hat{x}||^2. \tag{29}$$

By matching the variational density with the INR-based video coding framework, we observe that minimizing the KL divergence corresponds to optimizing weights for the rate–distortion performance.

## C  APPROXIMATING HESSIAN

While MSE is the most commonly used loss function in INR-VC, it is impractical to assume that all future scenarios encountered by NeuroQuant will use MSE as their objective. To address this limitation, we consider an alternative approach based on the variation inference discussed earlier. The loss function of INR-VC can be framed as the likelihood estimation for $p(x|\tilde{w})$, allowing us to define the Hessian as:

$$\boldsymbol{H} = \nabla_{\tilde{w}}^2 \mathcal{L} = \mathbb{E}_{x\sim p(x)}\left[-\nabla_{\tilde{w}}^2 \log p(x|\tilde{w})\right]. \tag{30}$$

The negative expected Hessian of the log-likelihood function is equivalent to the Fisher information matrix (FIM) (LeCun et al., 2002). We define the FIM, denoted by $\boldsymbol{F}$, as follows:

$$\boldsymbol{F} = \mathbb{E}_{x\sim p(x|\tilde{w})}\left[-\nabla_{\tilde{w}}^2 \log p(x|\tilde{w})\right] \tag{31}$$

$$= \mathbb{E}_{x\sim p(x|\tilde{w})}\left[-\frac{1}{p(x|\tilde{w})}\nabla_{\tilde{w}}^2 p(x|\tilde{w})\right] + \mathbb{E}_{x\sim p(x|\tilde{w})}\left[\nabla_{\tilde{w}} \log p(x|\tilde{w})\nabla_{\tilde{w}} \log p(x|\tilde{w})^T\right]. \tag{32}$$

The first term on the right side of Eq. 32 is zero, as shown in Chen et al. (2021b):

$$\mathbb{E}_{x\sim p(x|\tilde{w})}\left[-\frac{1}{p(x|\tilde{w})}\nabla_{\tilde{w}}^2 p(x|\tilde{w})\right] \tag{33}$$

$$= \int \frac{1}{p(x|\tilde{w})}\nabla_{\tilde{w}}^2 p(x|\tilde{w})p(x|\tilde{w})dx \tag{34}$$

$$= \int \nabla_{\tilde{w}}^2 p(x|\tilde{w})dx \tag{35}$$

$$= \nabla_{\tilde{w}}^2 \int p(x|\tilde{w})dx \tag{36}$$

$$= 0. \tag{37}$$

Therefore,

$$\boldsymbol{F} = \mathbb{E}_{x\sim p(x|\tilde{w})}\left[-\nabla_{\tilde{w}}^2 \log p(x|\tilde{w})\right] \tag{38}$$

$$= \mathbb{E}_{x\sim p(x|\tilde{w})}\left[\nabla_{\tilde{w}} \log p(x|\tilde{w})\nabla_{\tilde{w}} \log p(x|\tilde{w})^T\right]. \tag{39}$$

In Eq. 11, the first term is ignored, and we approximate $\boldsymbol{H}$ using Gauss-Newton matrix form $\boldsymbol{G}$. Similarly, in the context of likelihood estimation, this can be represented as:

$$\boldsymbol{G} = \mathbb{E}_{x\sim p(x)}\left[\nabla_{\tilde{w}} \log p(x|\tilde{w})\nabla_{\tilde{w}} \log p(x|\tilde{w})^T\right]. \tag{40}$$

To summarize, we have the following relationships:

$$\boldsymbol{H} = \nabla_{\tilde{w}}^2 \mathcal{L} = \mathbb{E}_{x\sim p(x)}\left[-\nabla_{\tilde{w}}^2 \log p(x|\tilde{w})\right], \tag{41}$$

$$\boldsymbol{G} = \mathbb{E}_{x\sim p(x)}\left[\nabla_{\tilde{w}} \log p(x|\tilde{w})\nabla_{\tilde{w}} \log p(x|\tilde{w})^T\right], \tag{42}$$

$$\boldsymbol{F} = \mathbb{E}_{x\sim p(x|\tilde{w})}\left[-\nabla_{\tilde{w}}^2 \log p(x|\tilde{w})\right] \tag{43}$$

$$= \mathbb{E}_{x\sim p(x|\tilde{w})}\left[\nabla_{\tilde{w}} \log p(x|\tilde{w})\nabla_{\tilde{w}} \log p(x|\tilde{w})^T\right]. \tag{44}$$

In INR-VC, the observed data is consistent with the true data. Hence, when the target data distribution $p(\boldsymbol{x})$ equals to the fitted distribution $p(\boldsymbol{x}|\tilde{\boldsymbol{w}})$, we derive:

$$\boldsymbol{H} = \boldsymbol{G} = \boldsymbol{F}. \tag{45}$$

This approximation, constrained by network capacity (e.g., bitrate) without data distribution, reveals a unique advantage compared to generalized codecs. Therefore, we can get an equivalent objective under the FIM form, though not exactly the same as the Hessian form

$$\min \mathbb{E}\left[\Delta \boldsymbol{z}^T diag\left((\frac{\partial \mathcal{L}}{\partial \boldsymbol{z}_1})^2, \cdots, (\frac{\partial \mathcal{L}}{\partial \boldsymbol{z}_n})^2\right)\Delta \boldsymbol{z}\right] \tag{46}$$

For some loss functions, such as perceptual loss (Rad et al., 2019), which lack an analytical Hessian form, the FIM can serve as a suitable subsitute.

## D NEUROQUANT ALGORITHM

In our approach, $s$ represents the quantization steps. Once each layer is assigned a bitwidth $b^l$, the initial value of $s$ is computed channel-wisely using

$$s^{l,k} = \frac{\max(w^{l,k}) - \min(x^{l,k})}{2^{b^l} - 1}, \tag{47}$$

where $w^{l,k}$ is the weights in the $k$-th channel of the $l$-th layer. During the calibration process, $s$ is further optimized to minimize task loss.

---

**Algorithm 1** Bit Allocation for Target Bitrate (Mixed Precision, Sec. 3.1)

---

**Input:** Pretrained weights $\boldsymbol{W}$ (FP32); Initial bitwidth $b$ (e.g., 8 bit); Potential mixed-precision configures set $\boldsymbol{S}$ under target rate $R$
**Output:** Quantization steps $s$, Quantized weights $\widehat{\boldsymbol{W}}$
 1: Quantize $\boldsymbol{W}$ to initial bitwidth $b$ and get $\boldsymbol{W}_0$;
 2: Compute the gradient of $\boldsymbol{W}_0$ with backward propagation $\boldsymbol{g} = \frac{\partial \mathcal{L}}{\partial \boldsymbol{W}_0}$;
 3: **for** $S_i$ in $\boldsymbol{S}$ **do**
 4:      Get layer-wise bitwidth $b^l$ from $S_i$;
 5:      **for** $l = 1, 2, \cdots, N$-th layer in $\boldsymbol{W}_0$ **do**
 6:          Compute channel-wise quantization parameter $\{s^{l,k}\}_{k=1}^{k=c}$;                  ▷ Eq. 47
 7:          Compute de-quantized $\widehat{\boldsymbol{W}}^l = \boldsymbol{s}^l \cdot Round(\boldsymbol{W}_0/\boldsymbol{s}^l)$;
 8:      **end for**
 9:      Compute weight perturbation $\Delta \boldsymbol{W} = \boldsymbol{W} - \widehat{\boldsymbol{W}}$;
10:      Compute Hessian-Vector Product $\boldsymbol{H}\Delta \boldsymbol{W}$ with $\nabla \mathcal{G} = \nabla(\boldsymbol{g}\Delta \boldsymbol{W})$;             ▷ Eq. 8
11:      Compute criteria $\Omega_i = \Delta \boldsymbol{W}^T \boldsymbol{H} \Delta \boldsymbol{W}$;                              ▷ Eq. 9
12: **end for**
13: $\boldsymbol{s}, \widehat{\boldsymbol{W}} = \arg\min \Omega$.

---

## E EXPERIMENTS

### E.1 IMPLEMENTATION DETAILS

**Evaluation.** We conducted experiments on the UVG dataset[2], which consists of 7 videos, each with a resolution of $1920 \times 1080$ and recorded at 120 FPS over 5 or 2.5 seconds. We applied a center crop to achieve a resolution of $1920 \times 960$, similar to the preprocessing used in HNeRV and NeRV. For evaluation metrics, we employed PSNR to measure reconstruction distortion and bits per pixel (bpp) to access bitrate. Additionaly, the Bjøntegaard Delta Rate (BD-Rate) (Bjontegaard, 2001) was calculated for each baseline codec.

---

[2]Beauty, Bosphorus, HoneyBee, Jockey, ReadySetGo, ShakeNDry, YachtRide

---

**Algorithm 2** Encoding for Target Rate (Calibration, Sec. 3.2)

---

**Input:** Pretrained weights $W$ (FP32); Potential mixed-precision configures set $S$ under target rate $R$; iteration T

**Output:** Bitstream, bpp

1: Search optimal bit configure and get $s, \widehat{W}$;                                       ▷ Algorithm 1
2: Forward propagation and collect the FP network output $z$;
3: **for** $i = 1, 2, \cdots, T$ **do**                                                          ▷ Calibration
4:     Forward propagation and collect the quantized network output $\hat{z}$;
5:     Compute $\mathcal{L}$ and gradient descent;                                              ▷ eq. 16
6:                  ▷ Update $s$ first with a few iteration and then update rounding parameters $v$
7: **end for**
8: Quantize $W$ to integer with calibrated QPs, i.e., $s, v$;
9: Get bitstream with lossless entropy coding.
10: Compute bpp                                                                                 ▷ eq. 48

---

Here, the bpp calculation includes both the quantized network parameters $\hat{w}$ and the quantization parameters $s$:

$$bpp = bpp_w + bpp_s = \frac{\sum E(\hat{w})}{H \times W \times T} + \frac{\sum s \cdot b_s}{H \times W \times T}, \tag{48}$$

where $E$ denotes lossless entropy coding. For example, on a 1080p video sequence with INT4 $\hat{w}$ and FP16 $s$, the bpp for HNeRV-3M is approximately: $bpp_w \approx 0.01, bpp_s \approx 0.00004$. As shown, the contribution of $s$ to the overall bpp is negligible, but it is still included in all calculations for fairness.

**Baselines.** All baselines were implemented using open-source codes, including NeRV (Chen et al., 2021a), HNeRV(Chen et al., 2023), and HiNeRV (Kwan et al., 2024a).

**NeuroQuant.** In the mixed-precision bit allocation, we targeted the final weights size instead of the bpp, as the actual bpp cannot be accurately estimated after entropy coding. Specifically, the bitwidth was constrained within the range of $[3bit, 8bit]$. The weights compression ratio (FP32 weights size/ quantized weights size) was maintained within an average range of $[4.5, 10]$, where different baselines and weights size had small difference. We calculated potential bitwidth configurations while allowing a $5\%$ error tolerance to avoid empty solution. In current implementation, we conducted search in a group of predefined bitwidth configures for a given target size, which can be finished in one minute. Integer Programming (Hubara et al., 2021) also can achieve same objective.

Once the bits are allocated, we employed the Adam optimizer (Kingma, 2014) to calibrate quantization parameters (e.g, quantization steps, weight rounding) to minimize distortion. For frame-wise INR-VC systems like NeRV and HNeRV, the batchsize was set to 2, while for patch-wise INR-VC systems like HiNeRV, the batchsize was set to 144. The learning rate was set to $3e-3$ with a cosine annealing strategy. QP were be optimized for $2.1 \times 10^4$ iterations, although most cases converged in fewer than $1.5 \times 10^4$ iterations. All experiments were conducted using Pytorch with Nvidia RTX 3090 GPUs.

Our code will be made open-source upon the release of the paper.

**Task-Oriented PTQs.** Benchmarks were based on open-source implmentations of AdaRound, BRECQ, QDrop, and RDOPTQ. These methods were reproduced for INR-VC systems, using the same batchsize and learning rate as NeuroQuant, which provided better results than naive learning rate. Each layer/block was optimized for $2.1 \times 10^4$ iterations.

**INR-Oriented QATs.** For FFNeRV, the QAT process is summarised as follows:

$$\textbf{Forward:}\quad \hat{w} = sign(w) \cdot \frac{\lfloor (2^b - 1) \cdot \tanh(|w|) \rfloor}{2^b - 1}, \tag{49}$$

$$\textbf{Backward:}\quad \frac{\partial \mathcal{L}}{\partial \hat{w}} \approx \frac{\partial \mathcal{L}}{\partial w}. \tag{50}$$

For HiNeRV, the QAT process is based on QuantNoise (Stock et al., 2021) but forbidding the naive Straight-Through Estimator (STE):

$$\textbf{Forward:} \quad \tilde{\boldsymbol{w}} = \boldsymbol{w} \cdot mask + \hat{\boldsymbol{w}} \cdot (1 - mask), \tag{51}$$

$$where \quad \hat{\boldsymbol{w}} = \lfloor \frac{\boldsymbol{w} \cdot (2^b - 1)}{2 \cdot \max(\boldsymbol{w})} \rceil \cdot \frac{2 \cdot \max(\boldsymbol{w})}{2^b - 1}, \tag{52}$$

$$\textbf{Backward:} \quad \frac{\partial \mathcal{L}}{\partial \tilde{\boldsymbol{w}}} \approx \frac{\partial \mathcal{L}}{\partial \boldsymbol{w}} \cdot mask, \tag{53}$$

$$\textbf{Inference:} \quad \hat{\boldsymbol{w}} = \lfloor \frac{\boldsymbol{w} \cdot (2^b - 1)}{2 \cdot \max(\boldsymbol{w})} \rceil \cdot \frac{2 \cdot \max(\boldsymbol{w})}{2^b - 1} \tag{54}$$

where $mask$ is a random binary tensor with the same shape as $\boldsymbol{w}$. This random dropping is also similar to QDrop.

**Generalized Neural Video Codecs.** We also included two representative generalized Neural Video Coding Systems, DCVC (Li et al., 2021a) and DCVC-DC (Li et al., 2023), to compare with the existing non-generalized INR-VC systems. Pretrained models were used to test the UVG dataset, where all frames of each video were evaluated. The group of pictures (GOP) was set to 32, consistent with other learned video coding methods.

**Generalized Traditional Codecs.** The command to encode using x264 in our paper is:

```
ffmpeg
-s {width} x {height}
-pix_fmt yuv444p10le
-framerate {frame rate}
-i {input yuv name}
-c:v libx264
-preset veryslow
-g 32
-qp {qp}
{bitstream file name}
```

The command to encode using x265 in our paper is:

```
ffmpeg
-s {width} x {height}
-pix_fmt yuv444p10le
-framerate {frame rate}
-i {input yuv name}
-c:v libx265
-preset veryslow
-x265-params
``qp= {qp}:keyint=32''
{bitstream file name}
```

The pre-process followed the suggestions in Sheng et al. (2022); Li et al. (2024), where we used BT.601 color range to convert between YUV and RGB.

### E.2 VARIABLE-RATE COMPARISON

Here we further compare NeuroQuant with two typical variable-rate techniques: (1) Neural Network Coding (NNC) techniques (Wiedemann et al., 2020) and Entropy Regularization (EM) techniques (Gomes et al., 2023; Kwan et al., 2024b; Zhang et al., 2021). NNC uses video codec to compress neural network, while EM introduces additional weight entropy regularization. We employ Deep-CABAC (Wiedemann et al., 2020) and Gomes et al. (2023) as the typical represented methods, respectively. We use the NeRV across three video sequences: Beauty, Jockey, and ReadySetGo to compare. The results are summarized in the following table:

As shown, NeuroQuant consistently outperforms DeepCABAC and Gomes et al. across different sequences and bitrates. These results demonstrate the effectiveness of our method in improving rate-distortion performance. For one training iteration, Gomes et al. (2023) involves additional overhead due to entropy estimation, resulting in approximately $\times 1.4$ encoding time compared to NeuroQuant.

Table 3: Variable-Rate comparison.

| Methods | Bpp | Beauty | Jockey | ReadyS | Avg. |
|---|---|---|---|---|---|
| Full Prec. (dB) | - | 33.08 | 31.15 | 24.36 | 29.53 |
| DeepCABAC (Wiedemann et al., 2020) | 0.016 | 32.98 | 30.94 | 24.24 | 29.39 |
| Gomes et al. (Gomes et al., 2023) | 0.016 | 32.91 | 30.66 | 23.92 | 29.16 |
| NeuroQuant (Ours) | 0.016 | 33.04 | 31.09 | 24.31 | 29.48 |
| DeepCABAC (Wiedemann et al., 2020) | 0.013 | 32.43 | 30.23 | 23.92 | 28.86 |
| Gomes et al. (Gomes et al., 2023) | 0.013 | 32.78 | 30.29 | 23.61 | 28.89 |
| NeuroQuant (Ours) | 0.013 | 32.97 | 30.96 | 24.18 | 29.37 |
| DeepCABAC (Wiedemann et al., 2020) | 0.011 | 31.59 | 28.70 | 22.85 | 27.71 |
| Gomes et al. (Gomes et al., 2023) | 0.011 | 32.63 | 29.89 | 23.26 | 28.59 |
| NeuroQuant (Ours) | 0.011 | 32.83 | 30.67 | 23.85 | 29.12 |

Besides, NeuroQuant achieves precise bitrate control by adjusting quantization parameters, as bitrate is directly proportional to the number of parameters and their bitwidth. In contrast, entropy-based methods can not directly estimate compressed bitrate from the Lagrangian multiplier $\lambda$. Instead, it requires multiple encoding runs to fit a mapping from $\lambda$ to bitrate. This mapping is sequence-dependent, reducing its universality and reliability.

Additionally, we acknowledge that the lossless entropy coding used in NeuroQuant currently is less advanced compared to CABAC-based techniques used in DeepCABAC. However, as a quantization method, NeuroQuant is compatible with various entropy coding techniques. In future work, we aim to incorporate CABAC and EM techniques into NeuroQuant as discussed in Sec. 3.3.

### E.3   DIVING INTO NEUROQUANT

To analyze the efficiency of NeuroQuant, we conducted a series of deeper studies on the three sequences (Beauty, Jockey, and ReadySetGo) using NeRV-3M and 4-bit precision. Below, we summarize our findings.

#### E.3.1   IS INTER-LAYER DEPENDENCE A GOOD PROPERTY FOR INR?

**Benefits of Inter-Layer Dependencies:** INR models are inherently non-generalized and tailored to represent specific video data. This specialization often leads to strong inter-layer dependencies, where the contribution of each layer to the overall representation is tightly coupled. For video representation, such strong coupling can reduce redundancy across the network, leading to better rate-distortion performance in INR-VCs. In contrast, excessive independence could indicate poor utilization of the network's capacity.

**Challenges of Inter-Layer Dependencies:** Strong dependencies complicate quantization, as perturbations in one layer can propagate across the network. This is where network-wise calibration, as proposed in NeuroQuant, becomes critical. Additionally, dependencies can limit scalability and robustness, as architectural modifications can disrupt the internal balance of the network.

**Simple Experiment:** For INR-VCs, the overfitting (dependence) degree increases with the training iterations growing. To measure dependence, we quantized the fourth layer using vanilla MinMax quantization without any calibration, and observed its impact on final loss. Results for varying training epochs are shown below:

The results indicate that stronger dependence means larger loss changes with quantization but provides better representation performance.

For future work, exploring a middle ground—where moderate independence is encouraged to balance representation fidelity and quantization robustness—might be an interesting direction.

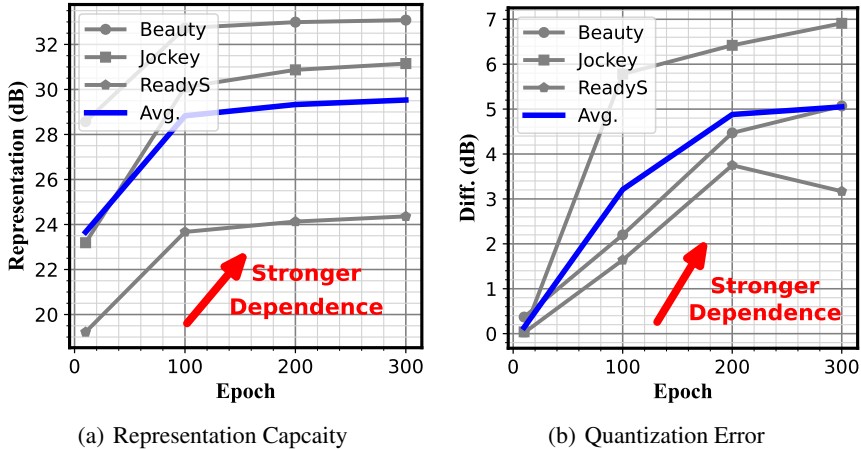

Figure 6: The influence of inter-layer dependence.

### E.3.2   ABLATION STUDIES

**Batch Size.** We examined the impact of varying batch sizes on compression performance. As shown in Table 4, increasing the batch size results in diminishing returns in PSNR, with larger batch sizes also introducing greater complexity. For NeuroQuant, a batch size of 2 was chosen as the default setting.

Table 4: Batch Size Ablation

|        | 1     | 2     | 4     | 8     |
|--------|-------|-------|-------|-------|
| Beauty | 32.78 | 32.79 | 32.79 | 32.80 |
| Jockey | 30.54 | 30.58 | 30.60 | 30.62 |
| ReadyS | 23.75 | 23.79 | 23.82 | 23.84 |
| Avg    | 29.02 | 29.05 | 29.07 | 29.09 |

**Learning Rate.** We experimented with different learning rates, as shown in Table 5. A learning rate of $3e-3$ was selected as the default NeuroQuant setting.

Table 5: Learning Rate Ablation

|        | $1.5e-3$ | $2e-3$ | $3e-3$ | $5e-3$ |
|--------|----------|--------|--------|--------|
| Beauty | 32.86    | 32.81  | 32.83  | 32.79  |
| Jockey | 30.58    | 30.62  | 30.67  | 30.53  |
| ReadyS | 23.88    | 23.82  | 23.85  | 23.79  |
| Avg    | 29.11    | 29.08  | **29.12** | 29.04 |

**Objective Trade-off.** Eq. 14 describes the final calibration objective, which includes both the distortion term $\mathcal{L}_D$ and the regularization term $\mathcal{L}_{Reg}$. We explored the trade-off between these two objectives by varying the regularization weight $\lambda$, as shown in Table 6. The results indicate that NeuroQuant maintains robust performance across different regularization weights.

**Quantization Granularity.** We compared different quantization granularities, focusing on channel-wise (CW) and layer-wise (LW) quantization strategies. As shown in Table 7, while layer-wise granularity significantly degrades performance, NeuroQuant still yields satisfactory results in this context. Channel-wise quantization delivers superior performance, supporting the correctness of the proposed method.

Table 6: Objective Trade-off Ablation

|        | $\lambda = 0.1$ | $\lambda = 0.01$ | $\lambda = 0.001$ |
|--------|-----------------|------------------|-------------------|
| Beauty | 32.79           | 32.78            | 32.78             |
| Jockey | 30.58           | 30.58            | 30.56             |
| Ready  | 23.80           | 23.77            | 23.76             |
| Avg    | 29.06           | 29.04            | 29.03             |

Table 7: Quantization Granularity Ablation

|        | LW w/o NeuroQuant | LW w/ NeuroQuant | CW w/ NeuroQuant |
|--------|-------------------|------------------|------------------|
| Beauty | 28.90             | 32.12            | 32.79            |
| Jockey | 22.49             | 29.08            | 30.58            |
| ReadyS | 19.50             | 22.51            | 23.80            |
| Avg    | 23.63             | 27.90            | 29.06            |

**Iterations.** We analyzed the performance gains over different numbers of iterations, as illustrated in Fig. 7. NeuroQuant achieves significant improvements in the first $10^3$ iterations, indicating its rapid convergence during calibration.

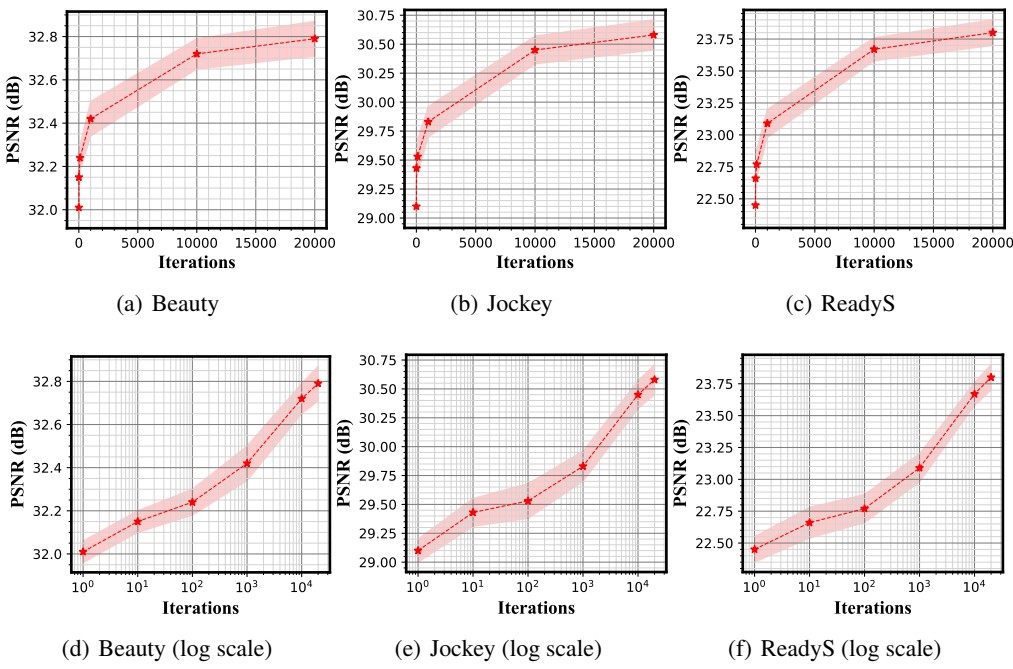

Figure 7: Iterations Ablation.

Due to the tremendous encoding complexity of HiNeRV ($10\times$ compared to NeRV, e.g., more than 21GB memory requirement and nearly one GPU day for 3M weights), we exclude HiNeRV in next.

**Unified R-D Characteristics.** Fig. 8 presents the Rate-Distortion (R-D) curves for three different video sequences: Jockey, Beauty, and ReadySetGo. In addition to the commonly used Jockey sequence, we include the less dynamic Beauty sequence, where baseline models tend to exhibit saturation (showing less than 1 dB PSNR improvement across the entire bitrate range). We also consider the highly dynamic ReadySetGo sequence, where all baselines show the worst R-D performance within the UVG dataset.

NeuroQuant significantly outperforms naive quantization techniques in terms of compression efficiency. For the Beauty sequence, both NeRV and HNeRV show a saturation effect, where increasing the number of weights does not yield noticeable gains in distortion reduction. In this case, NeuroQuant performs worse compared to its performance on other sequences..

On the other hand, NeuroQuant exhibits slight difference among different beseliens and video sequences. As highlighted in the zoomed-in areas of the R-D curves, *NeuroQuant-4bit* achieves higher PSNR in the *NeRV on ReadySetGo*. When it comes to the *NeRV on Beauty*, *NeuroQuant-5bit* achieves higher PSNR. Despite these differences, various precision levels (bitwidth) display nearly unified R-D characteristics.

**Visualization.** We further visualize the lowest bitrate ($0.01bpp$) for all three sequences in Fig. 9, 10 and 11. Compared with naive 32bit floating-point HNeRV (FP32), NeuroQuant can compress weights to INT4 without visual information loss.

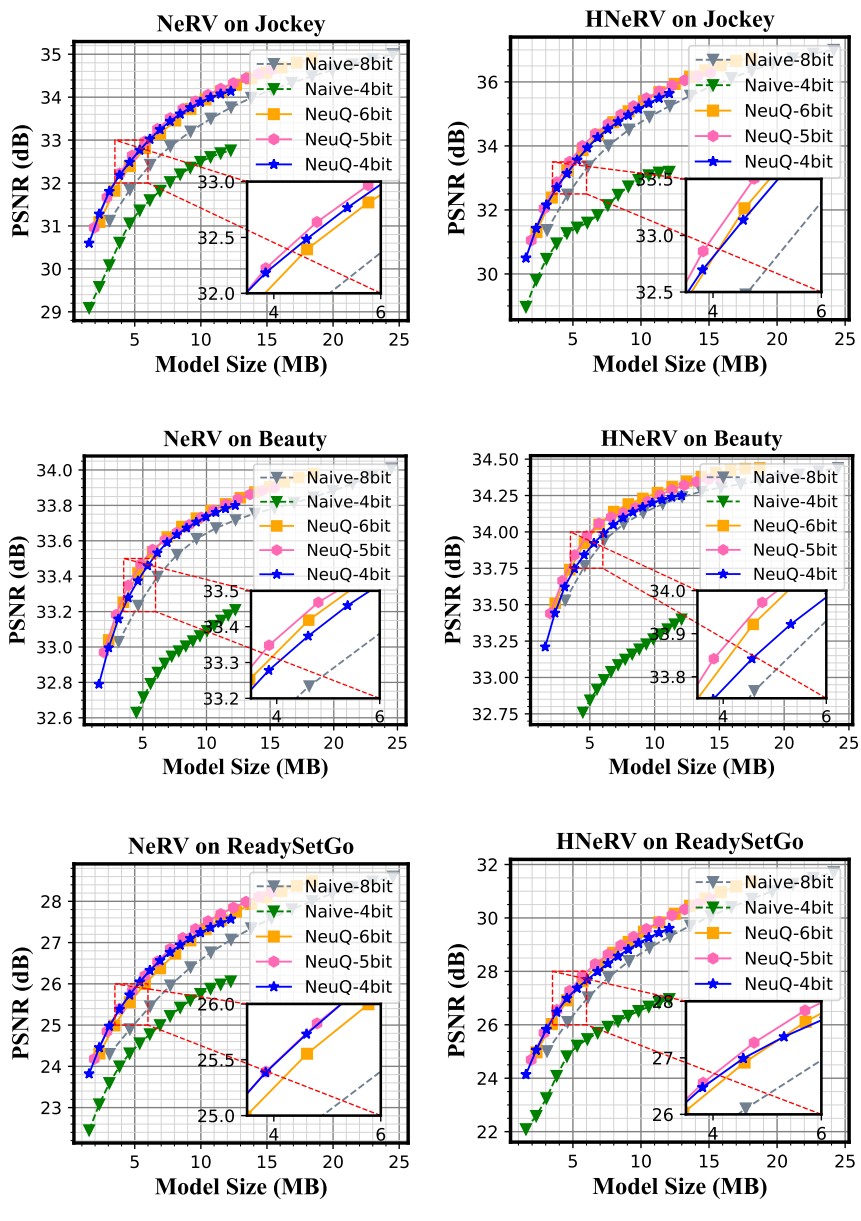

Figure 8: Quantitative comparison of NeRV (left) and HNeRV (right).

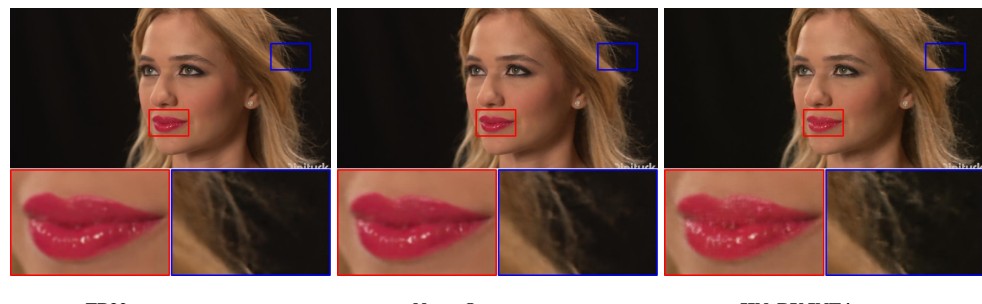

**FP32**
**PSNR: 33.33dB**
**Model Size: 12.4 MB**

**NeuroQuant**
**PSNR: 32.99dB**
**Model Size: 1.55 MB**

**HNeRV INT4**
**PSNR: 32.19dB**
**Model Size: 1.55 MB**

Figure 9: Comparison around $0.01bpp$ with Beauty sequence in HNeRV.

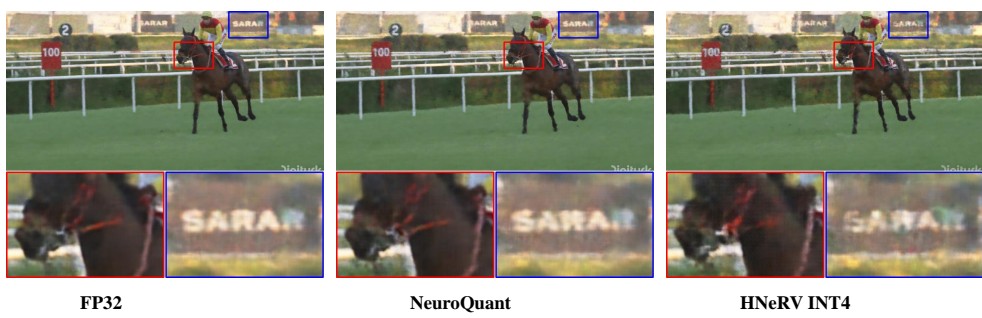

**FP32**
**PSNR: 29.73dB**
**Model Size: 12.4 MB**

**NeuroQuant**
**PSNR: 28.56dB**
**Model Size: 1.55 MB**

**HNeRV INT4**
**PSNR: 26.84dB**
**Model Size: 1.55 MB**

Figure 10: Comparison around $0.01bpp$ with Jockey sequence in HNeRV.

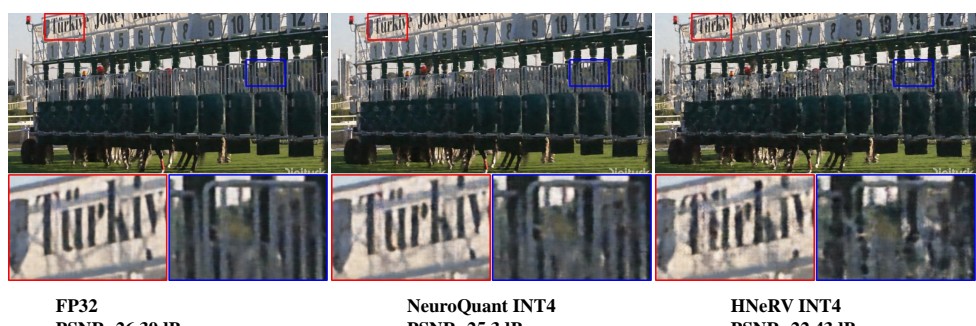

**FP32**
**PSNR: 26.39dB**
**Model Size: 12.4 MB**

**NeuroQuant INT4**
**PSNR: 25.3dB**
**Model Size: 1.55 MB**

**HNeRV INT4**
**PSNR: 22.43dB**
**Model Size: 1.55 MB**

Figure 11: Comparison around $0.01bpp$ with ReadySetGo sequence in HNeRV.

