# OpenReview forum: "On Quantizing Neural Representation for Variable-Rate Video Coding"
_ICLR.cc/2025/Conference — ICLR 2025 Spotlight_

### Official Review · Reviewer_6keJ · 2024-11-02

**Soundness:** 3
**Presentation:** 3
**Contribution:** 3
**Rating:** 8
**Confidence:** 4

**Summary:**

In this paper, the authors aim to introduce the variable rate control for INR-based video coding, by simply using PTQ. Therefore, they investigate PTQ approaches with mixed precision on those INR models. They first validate a weak layer independence in such non-generalized INR models. This challenges Hessian-based quantization methods, as they often follow this assumption and adopt diagonal Hessians. Then the authors propose a perturbation-based approach to estimate the intractable Hessian-involved sensitivity criterion (Omega) in the section with eq.9 and eq.10. Therefore, they can perform bit allocation for mix-precision quantization. Then the authors adopt network-wise calibration to further decrease the quantization error. The proposed approach named NeuroQuant achieves a cutting-edge performance w.r.t. both single QP and the whole RD curve.

**Strengths:**

The paper is well-written and easy to follow. The authors clearly explain their motivation for adopting mix-precision PTQ for variable-rate INR video coding. The experimental results are impressive, with significant PSNR improvements (*e.g.* 0.2 db @6bit and 3 db @2bit for NerV) on all the experimental settings.

**Weaknesses:**

1. The comparison in Table 1 may be unfair. If I understand correctly, the proposed approach involves mix-precision quantization, which helps bit allocation among layers. Therefore, it introduces extra quantization step parameters (${s}$ in eq.17) to store. I wonder whether the bpp calculation in Fig.4 and Fig.5 considers this quantization parameter. On the other hand, AdaQuant and BRECQ are fix-precision methods so this parameter can be omitted. The authors should clarify their evaluation details, especially the calculation of bpp.
1. The calibration objective derivation in section 3.2 is similar to the *Network-wise Reconstruction* situation discussed in the existing BRECQ paper (Li et al. 2021b, section 3.2). And the authors are also aware of this prior approach. Intuitively, intra-network independence can be seen as one-block intra-block independence, and BRECQ covers this. In behavior, both the calibration methods adopt an MSE-form objective. Thus, I cannot easily recognize those analyses as the contribution of this paper. I request the authors further clarify their contribution against the existing approaches.
2. It would be better to provide more intuitive explanations of the proposed approach, e.g. diagram figures and pseudo algorithm. Considering that not all experts in the video coding community are familiar with model quantization, the math formulations are somewhat confusing.

**Questions:**

1. I wonder why the improvement is such impressive, compared to the existing quantization approaches. Is the channel-wise quantization conducted on both weight and activation?
1. It seems like the key point of this approach is to adopt a mix-precision one-block BRECQ on INR video coding models. Despite the story about the unique properties of non-generalized INR models, can we further develop the method to a more general MP PTQ with calibration?
2. Is this inter-layer independence a good property for evaluated INR models, or an ill pose? I would appreciate if the authors provide some insight about this property.

---

> ### Author Response · Authors · 2024-11-21
> **Response to Reviewer 6keJ - Part I**
>
> Thank you for your insights and positive feedback on this paper. Below is the detailed response to each question and we hope we have addressed your concerns.
>
> >Q1: The authors should clarify their evaluation details, especially the calculation of bpp.
>
> A1: We apologize for the lack of clarity in our original description and provide the following clarifications:
>
> **Storage of $s$:** All quantization methods have to store the quantization step parameter $s$ because $s$ maps floating point weight $w$ to the quantized integer weight $\hat{w}$:
> \begin{align}
>         \hat{w} = round (\frac{w}{s}), \ \ \ s^{l,k} = \frac{\max (w^{l,k}) - \min (w^{l,k})}{2^{b^l} - 1},
> \end{align}
> with $w^{l,k}$ is the weights in the $k$-th channel of the $l$-th layer. During decoding, the original weight $w$ cannot be reconstructed from $\hat{w}$ without corresponding $s$. Even for fixed-precision methods, where the bitwidth $b^l$ is consistent across all layers, the weight distributions vary significantly among layers and channels (as shown in Fig. 3(a, b)). Therefore, $s$ must be stored for all methods, regardless of whether they use fixed or mixed precision.
>
> **Size of $s$:** In our implementation, we use channel-wise granularity for all compared methods, meaning all weights in a channel share the same $s$. Consequently, the number of stored $s$ depends solely on the number of channels. For example, in HNeRV (3M), the total number of weights and $s$ is 3M and 2.6K (less than 0.1\% of total weights), respectively.
>
> **Bpp Calculation:** The bpp calculation includes both the quantized network parameters $\hat{w}$ and the quantization parameters $s$:
> \begin{align}
>         bpp = bpp_w + bpp_s = \frac{\sum E(\hat{w})}{H\times W \times T} + \frac{\sum s \cdot b_s}{H\times W \times T},
> \end{align}
> where $E$ denotes lossless entropy coding. For example, on a 1080p video sequence with INT4 $\hat{w}$ and FP16 $s$, the bpp for HNeRV-3M is approximately: $bpp_w \approx 0.01, bpp_s \approx 0.00004$. As shown, the contribution of $s$ to the overall bpp is negligible, but it is still included in all calculations for fairness.
>
> We have detailed evaluation procedures in the updated manuscript.
>
> > Q2: I request the authors further clarify their contribution against the existing approaches.
>
> A2: We appreciate the opportunity to clarify the distinctions between NeuroQuant and existing methods like BRECQ. NeuroQuant introduces several novel contributions tailored specifically to the unique challenges of non-generalized INR-VCs:
>
> **Block vs. Network Independence:** We need to clarify that the "block" is concrete structure of a network. In INR-VCs, the "block" denotes a up-sampling block containing multiple layers. An INR-VC networks contains multiple up-sampling blocks.
>
> AdaRound first employs layer-wise calibration, while BRECQ observes that cross-layer dependencies primarily exist within blocks (e.g., residual bottleneck blocks) in generalized networks and performs block-wise calibration. This approach, however, does not hold for non-generalized INR-VCs. Through theoretical and experimental analysis, we demonstrate dependencies span the entire network (as illustrated in Fig. 3(c)), thereby leading to network-wise calibration. Such theoretical and experimental exploration for changing granularity from block to network is our key contribution. Neglecting network space structure and equating a network to a block will oversimplify the problems.
>
> **Difference behind MSE-form Objective:** NeuroQuant derives its MSE-form objective (Eq. 15) from the final MSE loss function used in video coding. In contrast, BRECQ’s MSE-form objective is derived from block-level Fisher approximations of the loss function (Eq. 10 in BRECQ) for block-wise calibration. The motivations and derivations differ significantly.
>
> **Channel-wise Quantization:** While AdaRound and BRECQ primarily employ layer-wise quantization, NeuroQuant introduces channel-wise quantization to better capture the unique weight distributions in INR-VCs. For fairness, we implement channel-wise quantization for all compared methods in our experiments. Without this adjustment, NeuroQuant would exhibit even larger performance gains.
>
> **Mixed Precision:** Another significant contribution of NeuroQuant is defining variable-rate coding as a mixed-precision bit allocation problem. We incorporate weight perturbation directionality and off-diagonal Hessian information to improve sensitivity assessment. Furthermore, we introduce the Hessian-Vector product to simplify computations, avoiding explicit Hessian calculations.
>
> **Better Performance:** NeuroQuant achieves substantial performance gains compared to BRECQ, validating the proposed mixed-precision and network-wise calibration approaches.
>
> These distinctions collectively form the foundation of NeuroQuant’s contributions. While we acknowledge BRECQ’s pioneering work, NeuroQuant extends the scope and applicability to address the unique challenges of INR-VCs.

---

> ### Author Response · Authors · 2024-11-21
> **Response to Reviewer 6keJ - Part II**
>
> >Q3: It would be better to provide more intuitive explanations of the proposed approach, e.g. diagram figures and pseudo algorithm.
>
> A3: Thank you for this thoughtful suggestion. We have included a pseudo-algorithm detailing the NeuroQuant pipeline in the updated Appendix D. Additionally, we plan to open-source our code upon publication of the manuscript to further aid reproducibility and understanding.
>
> >Q4: I wonder why the improvement is such impressive, compared to the existing quantization approaches. Is the channel-wise quantization conducted on both weight and activation?
>
> A4: Thank you for raising this important question. Below, we try to analyse their quantization approaches for inferior performance in INR-VCs.
>
> **Weight Activation:** The QAT technique in FFNeRV applies a weight activation function:
> \begin{align}
>     \text{Forward:}  \ \ \ \hat{w} = sign(w) \cdot \frac{\lfloor (2^b-1) \cdot \tanh(|w|) \rfloor}{2^b-1}.
> \end{align}
> Here, weights are constrained with $(-1,1)$ using $tanh(|w|)$. While this simplifies quantization, it also limits the network’s representational capacity, leading to degraded performance. In contrast, other methods avoid weight activation, preserving the better representational capacity of the network.
>
> **Straight-Through Estimator (STE):** Current QAT techniques rely on STE for backward due to in-differentiable round function, e.g., in FFNeRV:
> \begin{align}
>     \text{Backward:} \ \ \ \frac{\partial \mathcal{L}}{\partial \hat{w}} \approx \frac{\partial \mathcal{L}}{\partial w}.
> \end{align}
>
> However, STE leads to biasd gradients, hindering the accurate optimization. HiNeRV attempts to mitigate this issue by introducing a random binary mask:
>     \begin{align}
>         \text{Forward:}  \ \ \ & \tilde{w} = w \cdot mask + \hat{w} \cdot (1-mask), \\ \ \ \ \
>         \text{Backward:} \ \ \ & \frac{\partial \mathcal{L}}{\partial \tilde{w}} \approx \frac{\partial \mathcal{L}}{\partial w} \cdot mask,
>     \end{align}
> While this approach reduces the impact of quantized weights during training, it does not fully address STE’s inherent limitations. AdaRound, BRECQ and NeuroQuant employ differentiable round function combined with an annealing strategy, which allows the rounding operation to converge to the true rounding function over iterations. This avoids biased gradients and improves optimization accuracy. Similar STE-induced performance degradation has been observed in AdaRound and HiNeRV, with HiNeRV reporting a $0.02 - 0.48$ PSNR drop due to STE.
>
> **Layer/Block Calibration:** As shown in Fig. 3(c), INR-VCs exhibit significant cross-layer dependencies across the entire network. NeuroQuant employs network-wise calibration, capturing these dependencies more effectively than the layer-wise calibration in AdaRound or the block-wise calibration in BRECQ, QDrop, and RDOPTQ.
>
> **Mixed Precision:** NeuroQuant uses mixed precision for variable-rate coding, achieving superior bit allocation compared to uniform precision methods. Unlike HAWQ-v2 [1], which relies solely on diagonal Hessian information, NeuroQuant incorporates weight perturbation directionality and off-diagonal Hessian information for better sensitivity analysis (see left subfigure of Fig. 5).
>
>
> **Fair Comparison:** For fairness, all methods were implemented with channel-wise quantization. Original layer-wise quantization in methods like AdaRound and BRECQ would degrade performance significantly (e.g., $>1$ dB). From a transmission perspective, bitrate depends only on the weights, not activations. Therefore, activation quantization was not applied in NeuroQuant or any of the compared methods.

---

> ### Author Response · Authors · 2024-11-21
> **Response to Reviewer 6keJ - Part III**
>
> >Q5: It seems like the key point of this approach is to adopt a mix-precision one-block BRECQ on INR video coding models. Despite the story about the unique properties of non-generalized INR models, can we further develop the method to a more general MP PTQ with calibration?"
>
> A5: Thank you for this thoughtful question. Before addressing potential extensions, we have to emphasize, as discussed in Q2, that treating NeuroQuant as one-block BRECQ is not accurate just as we cannot treat BRECQ/QDrop as one-layer AdaRound. Below, we outline some potential pathways for extension:
>
> **Enhanced Sensitivity Analysis:** NeuroQuant’s mixed-precision criteria ($\Delta w^T H^{(w)} \Delta w $, Sec. 3.1) incorporates global Hessian information ($H^{(w)}$) and accounts for weight perturbation anisotropy $\Delta w$. While current generalized networks exhibit cross-layer dependencies primarily within blocks (e.g., Residual Blocks), considering global dependencies can provide more accurate second-order loss estimations. Besides, NeuroQuant simplifies Hessian calculating by introducing the Hessian-Vector product. This makes our MP adaptable to various networks. For deeper networks with exponentially growing bitwidth configurations, fast search strategies like integer programming [2] or genetic algorithms [3] could be employed.
>
> **Broadening Calibration Granularity:** NeuroQuant employs network-wide calibration for INR-VCs, while generalized networks prefer block-wise calibration. To evaluate this, we applied network-wise calibration to generalized codecs using the open-source RDOPTQ. Results, measured in BD-Rate, are shown below:
> | RDO-PTQ                  | W-bit | Kodak  | Tecnick | CLIC   |
> |-------------------------|-------|--------|----------|---------|
> | block calib.   | 6      | 7.92%  | 12.02%  | 11.28% |
> | network calib. | 6      | 9.25%  | 14.36%  | 12.50% |
> | block calib.   | 8      | 0.82%  | 2.02%   | 1.87%  |
> | network calib. | 8      | 1.15%  | 2.41%   | 2.03%  |
>
> These results suggest that plain network calibration is not optimal for generalized networks. A potential strategy is to introduce a modular granularity approach, allowing transition between layer-, block-, and network-wise calibration, depending on the specific architecture and use case.
>
> We appreciate your novel perspective and will further explore these possibilities in future work.
>
> >Q6: Is this inter-layer independence a good property for evaluated INR models, or an ill pose? I would appreciate if the authors provide some insight about this property.
>
> A6: Thank you for this thought-provoking question. The inter-layer dependence in INRs is a double-edged sword, with both benefits and challenges:
>
> **Benefits:** INRs are inherently non-generalized and tailored to represent specific video. This often leads to strong inter-layer dependencies, where the contribution of each layer to the overall representation is tightly coupled. Such strong coupling can reduce redundancy across the network, leading to better representation performance. In contrast, excessive independence could indicate poor utilization of the network's capacity.
>
> **Challenges:** Strong dependencies complicate quantization, as perturbations in one layer can propagate across the network. This is where proposed network-wise calibration becomes critical. Additionally, dependencies can limit scalability and robustness, as architectural modifications can disrupt the internal balance of the network.
>
> **Experiment:** For INRs, the dependence degree increases with the training iterations growing. To measure dependence, we quantized the fourth layer using vanilla MinMax quantization without any calibration in three video sequence:
>
> | epoch       | 10    | 100   | 200   | 300   |
> |------------|-------|-------|-------|-------|
> | FP 32    | 23.66 | 28.83  | 29.33  | 29.53 |
> | INT 3    | 23.52 | 25.63  | 24.45  | 24.48 |
> | Diff.      | 0.14 | 3.21  | 4.88  | 5.05  |
>
> More results can be found in Appendix E. The results indicate that stronger dependence means larger quantization error but provides better representation performance. For future work, exploring a middle ground—where moderate independence is encouraged to balance representation fidelity and quantization robustness—might be an interesting direction.
>
> We appreciate your novel perspective, which has sparked meaningful discussions. We believe these discussions not only enhance our manuscript but also benefit practitioners in understanding the unique characteristics of non-generalized INRs.
>
> Thank you again for your time, effort, and valuable feedback in reviewing our manuscript.
>
> **Ref:**
>
> 1. Dong, Zhen, et al. "Hawq-v2: Hessian aware trace-weighted quantization of neural networks."
>
> 2. Hubara, Itay, et al. "Improving post training neural quantization: Layer-wise calibration and integer programming."
>
> 3. Guo, Zichao, et al. "Single path one-shot neural architecture search with uniform sampling."

---

> ### Comment · Reviewer_6keJ · 2024-11-25
> **Reviewer 6keJ feedback to author rebuttal**
>
> Thanks for the authors' detailed response. They address my concerns, post supplementary results, and provide more insights to demonstrate their claims in this paper. After thoroughly reading all the reviews and author responses, I believe this paper is solid to push the study of INR-based video coding. I raise my rating to acceptance.

---

> > ### Author Response · Authors · 2024-11-26
> >
> > We sincerely thank the reviewer. We learned a lot from the suggestions which helped us greatly improve our manuscript.

---

### Official Review · Reviewer_i6EH · 2024-11-03

**Soundness:** 3
**Presentation:** 3
**Contribution:** 3
**Rating:** 8
**Confidence:** 4

**Summary:**

This paper proposes a novel post-training quantization approach designed for INR-VC called NeuroQuant that enables variable-rate coding without complex retraining. It redefines variable-rate coding as a mixed-precision quantization problem. Through network-wise calibration and channel-wise quantization strategies, NeuroQuant achieves SOTA performance compared to popular PTQ and QAT methods.

**Strengths:**

Compared with existing quantization methods, the proposed method could achieve significant performance improvement, indicating the efficiency of the proposed method.
There are some highlights for the proposed post-training quantization method for INR-VC:
1.	A criterion for optimal sensitivity in mixed-precision INR-VC was proposed, enabling the allocation of different bitwidth to network parameters with varying sensitivities.
2.	Through network-wise calibration and channel-wise quantization strategies,  NeuroQuant minimize quantization-induced errors, arriving at a unified formula for representation-oriented PTQ calibration.

**Weaknesses:**

1.	The authors did not provide a detailed explanation as to why the proposed PTQ method would be superior to QAT methods such as FFNeRV and HiNeRV.
2.	In the Encoding Complexity section, the authors did not provide a detailed explanation of whether the acceleration brought by NeuroQuant is due to the absence of QAT optimization during training or because NeuroQuant does not require retraining for adjustments at different bitrates.

**Questions:**

1.	I think that the RD performance of the PTQ method may be slightly inferior to QAT. Could you explain why the proposed PTQ method has better RD performance compared to FFNeRV/HiNeRV?
2.	Could you provide a detailed explanation of the Encoding Complexity section? Does it refer to the encoding complexity of a single bitrate or multiple bitrate?

---

> ### Author Response · Authors · 2024-11-21
> **Response to Reviewer i6EH**
>
> We sincerely thank the reviewer for their constructive comments and are delighted that they appreciate the methodology and state-of-the-art performance of NeuroQuant. Below, we address the raised concerns in detail:
>
> >Q1 & Q3: Could you explain why the proposed PTQ method has better RD performance compared to FFNeRV/HiNeRV?"
>
> A1 & A3: Thank you for raising this important point. There are some potential factors for inferior performance of QATs.
>
> **Weight Activation:** The QAT method in FFNeRV applies a weight activation function:
> \begin{align}
>     \text{Forward:} \ \ \ \hat{w} = sign(w) \cdot \frac{\lfloor (2^b-1) \cdot \tanh(|w|) \rfloor}{2^b-1},
> \end{align}
> where weights are constrained within $(-1, 1)$ using $tanh(|w|)$. While this simplifies the quantization process, it also limits the network's capacity, potentially degrading performance. In contrast, other methods avoid weight activation, preserving the better representational capacity of the network.
>
> **Straight-Through Estimator (STE):**  Current QAT techniques rely on STE to approximate gradients for the non-differentiable round function:
> \begin{align}
>     \text{Backward:} \ \ \ \frac{\partial \mathcal{L}}{\partial \hat{w}} \approx \frac{\partial \mathcal{L}}{\partial w}.
> \end{align}
>
> However, STE introduces biased gradients, hindering accurate optimization. HiNeRV observed similar performance degradation and replaced STE with:
> \begin{align}
>         \text{Forward:}  \ \ \ & \tilde{w} = w \cdot mask + \hat{w} \cdot (1-mask), \\ \ \ \  \ \
>         \text{Backward:} \ \ \ & \frac{\partial \mathcal{L}}{\partial \tilde{w}} \approx \frac{\partial \mathcal{L}}{\partial w} \cdot mask,
> \end{align}
> where a random binary mask limits biased gradient propagation. While this mitigates STE’s issues, it does not address the fundamental problem.
>
> NeuroQuant, in contrast, uses a differentiable round function and an annealing strategy, allowing the round function to converge to the true round operation over iterations. Such differentiable round operations have been widely used in PTQs, which avoid biased gradients and improves optimization accuracy. This improves gradient accuracy and reduces performance degradation, as also observed in AdaRound and HiNeRV (e.g., STE caused a $0.02 - 0.48$ dB PSNR drop in HiNeRV).
>
> **Mixed Precision and Network-wise Calibration:** NeuroQuant leverages mixed precision to allocate bitwidths optimally across layers, improving efficiency and quality. The 2-bit results in Table 2 demonstrate NeuroQuant’s superior performance. The left subfigure of Fig. 5 directly compare our method with uniform quantization, and HAWQ-v2 [1], a leading mixed-precision method. Additionally, NeuroQuant’s network-wise calibration further enhances RD performance by accounting for inter-layer dependencies.
>
> >Q2 & Q4: Could you provide a detailed explanation of the Encoding Complexity section? Does it refer to the encoding complexity of a single bitrate or multiple bitrate?
>
> A2 & A4: We apologize for any confusion caused by the original description.
>
> For a given target bitrate, NeuroQuant performs a lightweight QPs calibration process based on pretrained weights, which requires only a few iterations. In contrast, naive NeRV, HNeRV, and HiNeRV adjust bitrates by modifying the number of weights, requiring separate models to be trained from scratch for each target bitrate, which is significantly more time-consuming.
>
> **Encoding Time of Table 2:** The encoding complexity reported in Table 2 refers to the time required to support a new bitrate. Since NeuroQuant shares a pretrained model across bitrates, its pretraining time was originally excluded. However, we acknowledge that this could be misleading and have updated Table 2 to include pretraining time for clarity.
>
> We hope these information is helpful and enhances your confidence in assessing our manuscript. Thank you again for your constructive comments and support.
>
> **Ref:**
> 1. Dong, Zhen, et al. "Hawq-v2: Hessian aware trace-weighted quantization of neural networks." Advances in neural information processing systems 33 (2020): 18518-18529.

---

### Official Review · Reviewer_3W48 · 2024-11-04

**Soundness:** 3
**Presentation:** 3
**Contribution:** 3
**Rating:** 6
**Confidence:** 4

**Summary:**

In this paper, the authors propose post-training quantization for INR-VCs, which achieves variable-rate coding more efficiently than existing methods that require retraining the model from scratch. The proposed model realizes variable bitrate by considering the sensitivity of the weights to quantization, while also incorporating better theoretical assumptions for INR-VC compared to other post-training quantization techniques. The proposed method demonstrates both improved RD performance and faster encoding speed.

**Strengths:**

- The results look promising. Although NeuroQuant achieves only a marginal improvement over the current best INR-VC (-4.8%), it provides greater efficiency in obtaining multiple rate points.
- The experiments comparing different quantization methods are comprehensive, which will be helpful for future work in this area.

**Weaknesses:**

- In Table 2, excluding the pretraining time for NeuroQuant does not seem appropriate. Even with NeuroQuant, pretraining is still required, and the current presentation may be misleading. The authors should consider reporting the pretraining and fine-tuning times separately for both the baseline models and NeuroQuant.
- Similarly, the claim of an 8x encoding speedup is also misleading, as it excludes the pretraining time required for INR-VC encoding (even though NeuroQuant avoids full retraining for each rate point).
- Variable/learnable quantization levels for INR-VC have been explored in related works [1,2,3], so the paper’s claim is inaccurate (e.g., line 43). These methods, which resemble the proposed mixed-precision quantization, also enable fine-tuning for multiple rate points from a single pretrained model (but with QAT). These methods should be discussed and compared in the paper.
- For a fairer comparison, the comparisons to x264/x265 should avoid using the 'zerolatency' setting, as the INR-VCs in the paper inherently have non-zero latency.
- For ablation study, more sequences should be use for obtaining a representative result.

[1] Zhang, Yunfan, et al. "Implicit neural video compression."
[2] Gomes, Carlos, et al. "Video compression with entropy-constrained neural representations."
[3] Kwan, Ho Man, et al. "Immersive Video Compression using Implicit Neural Representations."

**Questions:**

- The encoding runtime for HiNeRV 3M (22 hours) looks significantly longer than reported in the original paper, even accounting for differences in GPUs. Additionally, the reported memory usage seems unusually high. Are there any differences in configuration, such as the use of FP16 precision in these experiments?
- How much time is required to obtain additional rate points with models like NeRV, HNeRV, FFNeRV, and HiNeRV? Although these models require a pretraining phase, the pretrained model can be fine-tuned to produce multiple rate points by adjusting quantization levels or using entropy coding with regularization [1]. Fine-tuning time is substantially shorter than full training (e.g., 30 epochs for QAT versus 300 + 60 epochs for HiNeRV).
- What is the computational cost (in terms of MACs and wall time) for the proposed calibration process compared to QAT?

[1] Gomes, Carlos, et al. "Video compression with entropy-constrained neural representations."

---

> ### Author Response · Authors · 2024-11-21
> **Response to Reviewer 3W48 - Part I**
>
> We sincerely thank the reviewer for their insightful feedback. Below, we clarify and address the concerns raised, and hope you can find them helpful:
>
> >Q1: The authors should consider reporting the pretraining and fine-tuning times separately for both the baseline models and NeuroQuant.
>
> A1: Thank you for this suggestion. In pretraining, all baseline models (e.g., NeRV, HNeRV, and HiNeRV) were trained for 300 epochs. In our implementation, all post-training quantization (PTQ) and fine-tuning techniques use the same pretrained model. Below, we report the pretraining and calibration times for NeuroQuant on an Nvidia RTX 3090 GPU using 32-bit floating-point (FP32) training precision:
>
> | Baselines   | Param. | Pretraining | Calibration for a bitrate |
> |-----------|--------|------------|----------------------|
> | NeRV      | 3.1 M  | 1.8 h     | 0.4 h                |
> | HNeRV     | 3.0 M  | 4.7 h     | 1.0 h                |
> | HiNeRV    | 3.1 M  | 18.9 h    | 2.8 h                |
>
> >Q2: Similarly, the claim of an 8x encoding speedup is also misleading, as it excludes the pretraining time required for INR-VC encoding (even though NeuroQuant avoids full retraining for each rate point).
>
> A2: Thank you for pointing this out. We have updated the manuscript to include pretraining time in Table 2.
>
> >Q3: Variable/learnable quantization levels for INR-VC have been explored in related works [1,2,3], so the paper’s claim is inaccurate (e.g., line 43). These methods should be discussed and compared in the paper.
>
> A3: Thank you for highlighting this. The related works [1, 2, 3] mentioned are entropy-based methods that use weight entropy regularization to constrain bitrate. Due to the limited rebuttal period, reproducing all three works was infeasible, especially since none of them are open-source. However, we try our best to reproduce [1] (Gomes et al.) as a representative entropy-based method. Our reproduced results are basically consistent with original [1] (NeRV-EM in original paper).
>
> Using the NeRV architecture as the baseline, we pretrained all models for 300 epochs. The comparison results are shown below:
>
> | Methods            | Bpp   | Beauty | Jockey | ReadyS | Avg.  |
> |-------------------|------|--------|---------|---------|-------|
> | Full Prec. (dB)   | -    | 33.08  | 31.15  | 24.36  | 29.53 |
> | Gomes.et al.  | 0.016 | 32.91  | 30.66  | 23.92  | 29.16 |
> | NeuroQuant        | 0.016 | 33.04  | 31.09  | 24.31  | 29.48 |
> | Gomes.et al.  | 0.013 | 32.78  | 30.29  | 23.61  | 28.89 |
> | NeuroQuant        | 0.013 | 32.97  | 30.96  | 24.18  | 29.37 |
> | Gomes.et al.  | 0.011 | 32.63  | 29.89  | 23.26  | 28.59 |
> | NeuroQuant        | 0.011 | 32.83  | 30.67  | 23.85  | 29.12 |
>
> The results demonstrate that NeuroQuant consistently outperforms Gomes et al. [1] across all sequences and bitrates. Discussions of these related works have been added to the Appendix E.2 of the revised manuscript and will be integrated into the main paper in the final version after peer review.
>
> >Q4: For a fairer comparison, the comparisons to x264/x265 should avoid using the 'zerolatency' setting, as the INR-VCs in the paper inherently have non-zero latency.
>
> A4: We apologize for the oversight in our execution of x264/x265 coding. The results in Figure 4 and the corresponding command details in Appendix E.1 have been updated without the "zerolatency" setting.
>
> >Q5: For ablation study, more sequences should be use for obtaining a representative result.
>
> A5:  Thank you for the suggestion. We have expanded our ablation study to include the Beauty and ReadySetGo sequences in addition to the original Jockey sequence. The updated results can be found in Appendix E.3. We appreciate your effort to enhance the clarity and comprehensiveness of the article.
>
>
>
> >Q6: Are there any differences in configuration, such as the use of FP16 precision in HiNeRV experiments?
>
> A6: To ensure fair and consistent comparisons across all baselines and benchmarks, we conducted all experiments in the same environment using FP32 training precision and PyTorch 1.10. The encoding runtime and memory usage reported in the paper reflect this uniform setup.
>
> For comparison, we further evaluated HiNeRV-3M using FP16 precision, as employed in the original HiNeRV paper. FP16 training was implemented using autocast and GradScaler from torch.cuda.amp. The results are summarized below:
> | Precision | HiNeRV | NeuroQuant |
> |-----------|---------|-----------|
> | FP32     | 22.2 h  | 2.8 h    |
> | FP16     | 15.0 h  | 1.8 h    |
>
> We found that FP16 precision significantly reduces encoding time for both naive HiNeRV and NeuroQuant, while maintaining performance comparable to FP32 (e.g., within $\pm 0.1$ dB). In the future, we plan to migrate all baselines and benchmarks to PyTorch 2.0 for consistency, as done in the original HiNeRV implementation.

---

> ### Author Response · Authors · 2024-11-21
> **Response to Reviewer 3W48 - Part II**
>
> >Q7: How much time is required to obtain additional rate points with models like NeRV, HNeRV, FFNeRV, and HiNeRV? Although these models require a pretraining phase, the pretrained model can be fine-tuned to produce multiple rate points by adjusting quantization levels or using entropy coding with regularization [1]. Fine-tuning time is substantially shorter than full training (e.g., 30 epochs for QAT versus 300 + 60 epochs for HiNeRV).
>
> A7: The time required to obtain a new bitrate from pretrained baselines (e.g., NeRV, HNeRV, and HiNeRV) is reported in the table provided in Q1, while reconstructed performance comparisons are shown in Q3. For comparison, we conducted 300 epochs of pretraining followed by 30 epochs finetuning for [1] as you suggested. Although both NeuroQuant and entropy-based methods [1,2,3] aim to achieve variable rates, NeuroQuant clearly outperforms these methods in several key aspects:
>
> **Better Reconstructed Quality:** For similar training iterations, NeuroQuant achieves better reconstructed quality, with an improvement of more than 0.32 dB over [1].
>
> **Lower Encoding Complexity:** For one training iteration, [1] involves additional overhead due to entropy estimation, resulting in approximately $\times 1.4$ encoding time compared to NeuroQuant.
>
> **Accurate Bitrate Control:** NeuroQuant achieves precise bitrate control by adjusting quantization parameters, as bitrate is directly proportional to the number of parameters and their bitwidth. In contrast, entropy-based methods [1,2,3] can not directly estimate compressed bitrate from the Lagrangian multiplier $\lambda$. Instead, it requires multiple encoding runs to fit a mapping from $\lambda$ to bitrate. This mapping is sequence-dependent, reducing its universality and reliability.
>
> Additionally, NeuroQuant is fundamentally a quantization method and is compatible with entropy regularization. As mentioned in Sec. 3.3, future work will explore the integration of NeuroQuant with entropy-based methods to further Rate-Distortion trade-off.
>
> >Q8: What is the computational cost (in terms of MACs and wall time) for the proposed calibration process compared to QAT?
>
> A8: Calibration and QAT are training processes for which exact MACs are difficult to measure. However, we highlight the runtime compared with current methods:
>
> **Compared with QAT in FFNeRV:** Due to the weight activation function $tanh(|w|)$, FFNeRV requires training from scratch. Without this, post-training weights suffer significant degradation when passed through $tanh$.
>
> **Compared with QAT in HiNeRV:** HiNeRV's QAT process has a runtime similar to NeuroQuant’s calibration process. However, naive HiNeRV requires training separate models from scratch for each potential bitrate. In this context, NeuroQuant provides a practical solution by leveraging a pretrained model, enabling adaptation to variable bitrates while ensuring efficient encoding.
>
> **Comparison with entropy-based Gomes et al. [1]:** As shown in Q3 and Q7, while [1] supports variable rates, it falls short in reconstructed quality, encoding complexity, and bitrate control accuracy compared to NeuroQuant.
>
> Overall, NeuroQuant demonstrates significant advantages in efficiency and practical applicability while maintaining high reconstruction quality.
>
> We have updated the manuscript to include additional details regarding encoding time, entropy-based variable bitrate methods, and extended ablation experiments. Thank you again for your valuable feedback and efforts in helping us improve the manuscript.
>
>
> **Ref:**
>
> 1. Gomes, Carlos, Roberto Azevedo, and Christopher Schroers. "Video compression with entropy-constrained neural representations." Proceedings of the IEEE/CVF Conference on Computer Vision and Pattern Recognition. 2023.
>
> 2. Zhang, Yunfan, et al. "Implicit neural video compression." arXiv preprint arXiv:2112.11312 (2021).
>
> 3. Kwan, Ho Man, et al. "Immersive Video Compression using Implicit Neural Representations." arXiv preprint arXiv:2402.01596 (2024).

---

> ### Comment · Reviewer_3W48 · 2024-11-26
>
> Thank you for the authors' response. The response and revised manuscript have addressed my concerns and questions.
> I will raise the rating accordingly.

---

> > ### Author Response · Authors · 2024-11-26
> >
> > Thank you again for your time, effort, and valuable feedback in reviewing our manuscript.

---

### Official Review · Reviewer_Zox4 · 2024-11-05

**Soundness:** 3
**Presentation:** 3
**Contribution:** 3
**Rating:** 6
**Confidence:** 3

**Summary:**

NeuroQuant is a cutting-edge post-training quantization method for variable-rate video coding that optimizes pre-trained neural networks for different bitrates without retraining.

**Strengths:**

1. The proposed method achieves variable-rate coding by adjusting QPs of pre-trained weights, eliminating the need for repeated model training for each target rate, which significantly reduces encoding time and complexity.

2. The method demonstrates superior performance in compression efficiency, outperforming competitors and enabling quantization down to INT2 without notable performance degradation.

3. The paper proposes a unified formula for representation-oriented PTQ calibration, streamlining the process and improving its applicability across different models.

4. The approach is backed by both empirical evidence and theoretical analysis, ensuring its robustness and effectiveness in practical applications.

**Weaknesses:**

N/A
Actually, I am not very familiar with this field, so please have AE consider the opinions of other reviewers more.

**Questions:**

How does NeuroQuant differ from traditional quantization methods in terms of bitrate adjustment?

---

> ### Author Response · Authors · 2024-11-21
> **Response to Reviewer Zox4**
>
> We would like to express our sincere gratitude for your thoughtful feedback and for taking the time to review our work. We greatly appreciate your recognition of our proposed method and the significance of our contributions. We are pleased to note that your understanding of our paper aligns well with the insights provided by other reviewers. Below, we address your question regarding bitrate adjustment:
>
> >Q1: How does NeuroQuant differ from traditional quantization methods in terms of bitrate adjustment?
>
> A1: Current video codecs can be broadly categorized into two types: (1) **Generalized codecs:** These methods used a generalized codec to extract features to various videos, including both handcrafted approaches (e.g., H.264 [1], H.265 [2]) and deep learning-based techniques (e.g., DCVC-DC [3]). (2) **Non-generalized INR-VCs:** These represent a specific video using a unique network, where video coding is transformed into encoding and transmitting network weights.
>
> **Compared with Quantization in Generalized Codecs:** Traditional quantization methods [3-5] often focus on quantizing feature maps (also named latents) to control bitrate, an approach is fundamentally different with weight quantization in INR-VCs. In contrast, NeuroQuant is the first to achieve bitrate control by adjusting the quantization parameters (QPs) of pretrained weights in non-generalized representations. Specifically, NeuroQuant introduces: (1) Fine-grained bitrate control by redefining quantization as a mixed-precision problem. (2) QPs optimization through network-wise calibration tailored for INR-VCs.
>
> **Compared with Weights Quantization:** NeuroQuant addresses two critical challenges when applying weight quantization to non-generalized INR-VCs: (1) Inter-layer Dependencies: Traditional methods assume inter-layer independence, which does not hold for INR-VCs. NeuroQuant incorporates weight perturbation directionality and off-diagonal Hessian information to address this. (2) Calibration Granularity: Existing layer/block calibration methods are ineffective for INR-VCs. NeuroQuant introduces network-wise calibration to optimize QPs.
>
> We apologize for the earlier omission of related background details. A discussion on bitrate adjustment has now been included in the Appendix A of the revised manuscript and will be incorporated into the final main paper after peer review.
>
> Your perspective is invaluable in deepening the understanding of our work. Thank you again for your positive feedback. We hope this information is helpful and enhances your confidence in assessing our contribution.
>
> **Ref:**
>
> 1. Wiegand, Thomas, et al. "Overview of the H. 264/AVC video coding standard." IEEE Transactions on circuits and systems for video technology 13.7 (2003): 560-576.
>
> 2. Sullivan, Gary J., et al. "Overview of the high efficiency video coding (HEVC) standard." IEEE Transactions on circuits and systems for video technology 22.12 (2012): 1649-1668.
>
> 3. Li, Jiahao, Bin Li, and Yan Lu. "Neural video compression with diverse contexts." Proceedings of the IEEE/CVF Conference on Computer Vision and Pattern Recognition. 2023.
>
> 4. Zhao, Tiesong, Zhou Wang, and Chang Wen Chen. "Adaptive quantization parameter cascading in HEVC hierarchical coding." IEEE Transactions on Image Processing 25.7 (2016): 2997-3009.
>
> 5. Wang, Hanli, and Sam Kwong. "Rate-distortion optimization of rate control for H. 264 with adaptive initial quantization parameter determination." IEEE Transactions on Circuits and Systems for Video Technology 18.1 (2008): 140-144.

---

### Official Review · Reviewer_M1Dk · 2024-11-08

**Soundness:** 3
**Presentation:** 3
**Contribution:** 3
**Rating:** 8
**Confidence:** 4

**Summary:**

In this paper, the authors propose a post-training quantization method tailored to implicit neural representation (INR) based image and video compression. They argue that existing post-training quantization methods are not suitable for INR-based image and video codecs, and advance the existing PTQ for this specific task. Furthermore, the authors demonstrate how their proposed method can tackle variable rate coding with INR using a single INR model. They experimented with their method on top of existing INR methods and showed that their method performs better with minimal reconstruction loss.

**Strengths:**

1) Using one single model for different bit rates with post-training quantization is interesting. This alleviates the need to train a model for each bit-rate, this will decrease the training time.

2) The paper provides the mathematical insights to their proposed method, inspired from the Nagel et. al (2020), and formulates the post-training quantization objective with respect to the network calibration.

3) The experimental results show that the proposed method has a significant gain in the variable-rate coding.

**Weaknesses:**

1. The authors failed to compare their proposed approach with Neural Network Coding tool (NNC) [1] which also performs post-training quantization, and also can offer variable-bitrate coding by adjusting QP parameters. The authors should compare their method with NNC.


 [1] S. Wiedemann et al., "DeepCABAC: A Universal Compression Algorithm for Deep Neural Networks," in IEEE Journal of Selected
          Topics in Signal Processing, vol. 14, no. 4, pp. 700-714, May 2020
          https://arxiv.org/abs/1907.11900

**Questions:**

1. How the figure 3 is generated. what is the architecture details of the INR network, and what kind of data is used for fitting. Does the analysis is also true for MLP?

2. What is the significance of the equation 5 and 6, whether this optimization problem is solved in the paper. In the abstract, it was mentioned, PTQ was formulated as the mixed-precision quantization, it was not evident for me where the mixed precision quantization is solved. From the table 1, it seems like the mixed-precision quantization was not used. Also detail how the mixed-precision quantization is used.

3.  For Nagel et.al (2020) which formulation was used to compare? In Nagel et. al (2020) whether the equation (25) or equation (21) is used in their respective paper. It is important to specify, the equation (25) is closer to the loss (network calibration) in the proposed paper.

4. In equation (16), how the $\mathbf{s}$ is determined, is it learned with respect to the task loss, or is it optimized by the greedy search or is it fixed parameter.

5.  the network-wise calibration might also be applicable to the generalized neural network, did the authors have done any experiments on the generalized neural codec?

6. For the quantization aware training approach, the weights initialization with post-training quantization will improve the convergence and reconstruction quality of the QAT. It would be nice to test this feature on some INR method which uses QAT.

---

> ### Author Response · Authors · 2024-11-21
> **Response to Reviewer M1Dk - Part I**
>
> We appreciate the reviewer’s thoughtful review and constructive feedback. Below is the detailed response to each question, hope you can find them helpful.
>
> >Q1: The authors should compare their method with NNC [1].
>
> A1: We appreciate the reviewer bringing up NNC [1]. We conducted a comparison between NeuroQuant and DeepCABAC [1], using the NeRV with 3.1M parameters across three video sequences. For DeepCABAC, different bitrates were achieved by adjusting the quantization step. The results are summarized in the following:
>
> | Methods       | Bpp  | Beauty | Jockey | ReadyS | Avg.  |
> |------------|------|--------|---------|---------|-------|
> | Full Prec. (dB) | -    | 33.08  | 31.15   | 24.36   | 29.53 |
> | DeepCABAC   | 0.016 | 32.98  | 30.94   | 24.24   | 29.39 |
> | NeuroQuant   | 0.016 | 33.04  | 31.09   | 24.31   | 29.48 |
> | DeepCABAC   | 0.013 | 32.43  | 30.23   | 23.92   | 28.86 |
> | NeuroQuant   | 0.013 | 32.97  | 30.96   | 24.18   | 29.37 |
> | DeepCABAC   | 0.011 | 31.59  | 28.70   | 22.85   | 27.71 |
> | NeuroQuant   | 0.011 | 32.8   | 30.67   | 23.85   | 29.12 |
>
> As shown, NeuroQuant consistently outperforms DeepCABAC across different sequences and bitrates. Additionally, the lossless entropy coding used in NeuroQuant currently is less advanced compared to CABAC in [1]. In future work, we aim to incorporate CABAC into NeuroQuant to further enhance performance.
>
> >Q2: How the figure 3 is generated.
>
> A2: We apologize for the lack of clarity in our explanation of Figure 3. Below, we provide more detailes:
>
> **Network and Data:**  All subfigures in Figure 3 are based on the HNeRV architecture with 1M parameters, and the Beauty sequence is used for fitting.
>
> **Subfigure (a) Layer statistics:**  We analyze the first convolution layer (blue) and the third convolution layer (orange). The histograms depict the relative number of parameters within narrow intervals, while the solid lines represent fitted probability distributions.
>
> **Subfigure (b) Channel's statistics:** We compute the maximum and minimum values across 84 channels in the fifth layer. This highlights the variability of weights among channels.
>
> **Subfigure (c) Hessian statistics:** The Hessian matrix $\mathbf{H}$ is derived as second-order information, where each entry $\mathbf{H}(i,j)$ represents $\frac{\partial \mathcal{L}^2}{\partial w_i \partial w_j}$, indicating the dependency between $i-th$ and $j-th$ weights. For HNeRV, we calculate the Hessian matrix across its 7 layers. In the visualization, we show the upper triangular Hessian matrix because it is nearly symmetric around the principal diagonal (the matrix is nearly Positive Semi-Definite (PSD) for a converged model).
>
> **Applicability to MLPs:** The analysis is fundamentally applicable to MLPs since the illustrated statistics are based on general principles of twice-differentiable neural networks, which is not constrained by the network's specific architecture. We will release the related code upon the acceptance of this paper.
>
> >Q3: What is the significance of the equation 5 and 6. Also detail how the mixed-precision quantization is used.
>
> A3: We appreciate the reviewer’s comments and provide the following clarifications:
>
> **Eq. 6** formulates the first key sub-problem of NeuroQuant: allocating bitwidths (i.e., mixed precision) to layers given a target bitrate $\mathcal{R} + \epsilon$. However, a specific bitrate can correspond to multiple mixed-precision configurations. Hence, it is crucial to determine the optimal allocation scheme. In **Sec. 3.1**, we analyze existing mixed-precision criteria and identify their shortcomings for non-generalized INRs due to significant inter-layer dependencies and weight perturbation directionality. We then propose a new criterion incorporating off-diagonal Hessian information and perturbation directionality.
>
> **Eq. 5** Once the bitwidths are allocated, the quantization parameters (QPs) can be initialized. Equation 5 describes the second sub-problem: calibrating initialized QPs to minimize the loss caused by quantization. This ensures better reconstruction quality. In **Sec. 3.2**, we derive a unified calibration formula based on equation 5, and introduce network-wise calibration and channel-wise quantization strategies.
>
> **Where is Mixed-Precision Result:** (1) Table 1: We first evaluate consistent bitwidths (e.g., INT6 and INT4). Mixed-precision results are included for INT2, marked with *. The results demonstrate NeuroQuant’s superiority across all configurations. (2) Sec. 4.2: The variable-rate coding experiments directly utilize mixed-precision quantization, showcasing its effectiveness in achieving better performance and reduced encoding time. (3) Fig. 5 (left subfigure): This directly compares our mixed-precision method against HAWQ-V2 [2], demonstrating the validity of our theoretical contributions.
>
> **Algorithm:** we provide a pseudo-algorithm detailing the mixed-precision process in the updated Appendix D.

---

> ### Author Response · Authors · 2024-11-21
> **Response to Reviewer M1Dk - Part II**
>
> >Q4: For Nagel et.al (2020) which formulation was used to compare?
>
> A4: The network calibration objective of NeuroQuant is described by the function $\mathcal{F}(x, w)$ in equation (16) of our manuscript, where $\mathcal{F}$ represents the networks's output for a given input $x$ and weights $w$.
>
> In contrast, AdaRound conducts layer-wise calibration, and $\mathcal{F}$ simplifies to the layer function (not activation function). Specifically, $\mathcal{F}(x, w)$ is equivalent to $f_a(wx)$ in equation (25) of AdaRound, where $f_a$ is the activation function. Therefore, in our implementation, AdaRound use equation (25) for layer calibration.
>
> Note that not all layers in various INR-VCs are followed by an activation function. For such cases, we employ equation (21) of AdaRound.
>
> >Q5: In equation (16), how the $s$ is determined?
>
> A5: In our approach, $s$ represents the quantization steps. Once the mixed precision scheme is determined, each layer is assigned a fixed bitwidth $b^l$. Then the initial value of $s$ is computed channel-wisely using
> \begin{align}
>         s^{l,k} = \frac{\max (w^{l,k}) - \min (x^{l,k})}{2^{b^l} - 1},
> \end{align}
> where $w^{l,k}$ is the weights in the $k$-th channel of the $l$-th layer.  During the calibration process, $s$ is further optimized to minimize task loss.
>
> >Q6: the network-wise calibration might also be applicable to the generalized neural network, did the authors have done any experiments on the generalized neural codec?
>
> A6: In generalized neural codecs, inter-layer dependencies are weaker due to their general-purpose representation. Furthermore, in the context of post-training quantization (PTQ), generalized codecs are more prone to overfitting when using small datasets for network-wise calibration, leading to higher testing errors.
>
> To explore the applicability of network-wise calibration to generalized codecs, we implemented the open-sourced RDO-PTQ, modifying its default layer/block-calibration approach to use network-wise calibration. We evaluated the generalized codec on three datasets using BD-Rate to jointly measure bitrate and PSNR, where the anchor is FP32 model:
> | RDO-PTQ                  | W-bit | Kodak  | Tecnick | CLIC  |
> |-------------------------|--------|--------|---------|--------|
> | block calib.   | 6      | 7.92%  | 12.02% | 11.28% |
> | network calib. | 6      | 9.25%  | 14.36% | 12.50% |
> | block calib.   | 8      | 0.82%  | 2.02%  | 1.87%  |
> | network calib. | 8      | 1.15%  | 2.41%  | 2.03%  |
>
> As shown, network-wise calibration performs worse than block calibration in generalized codecs. This suggests that plain network-wise calibration is not directly suitable for generalized codecs.  A potential strategy is to introduce a modular granularity approach, allowing the quantization to transition between layer-, block-, and network-wise calibration, depending on the specific models and use case.
>
> >Q7: For the quantization aware training approach, the weights initialization with post-training quantization will improve the convergence and reconstruction quality of the QAT. It would be nice to test this feature on some INR method which uses QAT.
>
> A7: Thank you for this valuable suggestion. We tested the impact of different PTQs for weight initialization in QAT from HiNeRV, using the NeRV (3.1M). The results for 4-bit quantization are presented below:
> | Method         | Beauty | Jockey | ReadyS | Avg.  |
> |--------------|--------|--------|--------|-------|
> | QAT          | 32.70  | 30.12  | 23.47  | 28.76 |
> | MinMax + QAT   | 32.55  | 29.65  | 23.22  | 28.47 |
> | MSE + QAT     | 32.64  | 29.93  | 23.30  | 28.62 |
> | DeepCABAC + QAT| 32.67  | 30.01  | 23.41  | 28.70 |
>
> Unfortunately, initializing weights with PTQ did not improve QAT for NeRV. This may be due to the unique characteristics of non-generalized INRs. Due to time constraints, we could not test additional sequences, QAT methods, or INR architectures. However, we recognize this as an important area of investigation and will explore other unique properties of non-generalized INRs further in future work.
>
> Due to the limited rebuttal period, we regret that we are unable to provide additional experimental results at this time. Following the peer review process, we will integrate these findings into the final version of the main paper. We believe that the novel views presented by you not only enhance the clarity of our manuscript but also draw attention from the research community to the unique characteristics and potential of non-generalized INRs.
>
> Thank you again for your time, effort, and valuable feedback in reviewing our manuscript.
>
> **Ref:**
>
> 1. S. Wiedemann et al., "DeepCABAC: A Universal Compression Algorithm for Deep Neural Networks," in IEEE Journal of Selected Topics in Signal Processing, vol. 14, no. 4, pp. 700-714, May 2020.
>
> 2. Dong, Zhen, et al. "Hawq-v2: Hessian aware trace-weighted quantization of neural networks." Advances in neural information processing systems 33 (2020): 18518-18529.

---

> ### Comment · Reviewer_M1Dk · 2024-11-26
>
> Thanks for addressing my concerns. The comparison with DeepCABAC is interesting. DeepCABAC determines the quantization step based on the reconstruction loss in the weights space which is different from NeuroQuant. If the DeepCABAC determines the quantization step based on the task loss, then the performance of the deepCABAC will be improved.

---

> > ### Author Response · Authors · 2024-11-27
> >
> > Thank you for your positive feedback and insightful observation.
> >
> > The key innovation of NeuroQuant lies in its **variable-rate coding** approach. Unlike the original DeepCABAC, which searches for optimal QPs (e.g., quantization step) by minimizing a rate-distortion (R-D) function for a given post-training model—resulting in a fixed bitrate. In our previous comparison (**Q1**), we adjusted DeepCABAC’s QPs to explore different R-D trade-offs (e.g., bpp vs. psnr in Q1) rather than searching for fixed QPs, aligning it with NeuroQuant’s variable-rate capability. Under variable-rate conditions, NeuroQuant consistently outperforms DeepCABAC across different sequences and bitrates.
> >
> >
> > When focusing solely on **optimal R-D coding without variable-rate adjustments**, we fully agree with your insights: determining QPs based on weight-space reconstruction loss is sub-optimal. Task-oriented QPs, as introduced by NeuroQuant, can yield superior performance.  A potential reason is that the weight-space loss does not equivalently reflect the task loss, whereas task-oriented QPs provide a more direct and effective optimization path.
> >
> > An exciting avenue for future research would involve combining the strengths of both approaches. While NeuroQuant excels in task-oriented QPs, DeepCABAC offers advanced CABAC-based entropy coding. Such an integration could achieve greater coding efficiency. However, a deeper exploration of DeepCABAC’s behavior and ablation studies are beyond the scope of the current manuscript but will be investigated in future work.
> >
> > We appreciate your novel perspective, which has sparked meaningful discussions. We hope this response addresses the concerns and  is helpful in assessing manuscript.

---

> > > ### Comment · Reviewer_M1Dk · 2024-11-27
> > >
> > > Thanks for the additional details, and including the details in the appendix of the manuscript. As the DeepCABAC has the closer performance with the NeuroQuant, and a simple method without any additional training, it is better to include them  somewhere in the main paper.
> > >
> > > Apart from this I don't have other concerns regarding the paper.

---

> > > > ### Author Response · Authors · 2024-11-28
> > > >
> > > > As you recommended, we have included DeepCABAC in the main paper, specifically in **Sec. 4.2** and **Fig. 4 (D-CABAC)**.
> > > >
> > > > We appreciate your time and efforts in enhancing the manuscript.

---

### Author Response · Authors · 2024-11-21
**Response to All**

We sincerely thank the reviewers for their thoughtful feedback and constructive suggestions. We are pleased that the reviewers recognized the innovation and state-of-the-art performance of the proposed NeuroQuant. Based on the reviewers’ comments, we have updated our manuscript to address their concerns and enhance clarity. The key updates include:

1. **Revised Statements, Figures, and Tables:** We have refined the presentation of statements, figures, and tables throughout the manuscript to improve clarity and avoid any potential misdirection.

2. **Background on Variable-Rate Coding (Appendix A):** We provide a discussion of related works on variable-rate coding for a better understanding of the context and contributions of NeuroQuant.


3. **Algorithm (Appendix D):** A pseudo algorithm is included to facilitate comprehension of NeuroQuant’s workflow.

4. **More Comparisons (Appendix E.2):** We present additional comparisons with Neural Network Coding and Entropy Regularization techniques, consistently demonstrating the superior performance of NeuroQuant.

5. **Expanded Ablation Studies (Appendix E.3):** Two additional video sequences have been included in the ablation studies to provide more comprehensive results. Also, we discuss the influence of inter-layer dependencies in INR models.

Additional explanations and results have been incorporated in the responses to each reviewers. After peer review, these revision and results will be incorporated into the final paper.

---

### Meta-Review · Area_Chair_GUDW · 2024-12-16

**Metareview:**

The paper was reviewed by five experts.

The authors provided responses, improved the manuscript and addressed the reviewers' raised concerns and convinced them to either maintain their initial positive ratings or augment them.

Following the rebuttal there is an unanimity -- all the reviewers are favorable to accepting the paper (3 x 8 and 2 x 6) and the AC recommends acceptance.

It is a solid work, a good paper, with significant contributions.

**Additional Comments On Reviewer Discussion:**

The authors did a fine job in improving the manuscript and addressing the reviewers' raised concerns and finally convince them and the AC on the significant contributions they make and results.

A couple of reviewers elevated their initial ratings.

---

### Decision · Program_Chairs · 2025-01-22

Accept (Spotlight)